# Testing an optimality-based model of rooting zone water storage capacity in temperate forests

Matthias J.R. Speich[1,2,3,a], Heike Lischke[1], Massimiliano Zappa[2]

[1]Dynamic Macroecology, Swiss Federal Research Institute WSL, 8903 Birmensdorf, Switzerland
[2]Hydrological Forecasts, Swiss Federal Research Institute WSL, 8903 Birmensdorf, Switzerland
[3]Department of Environmental Systems Science, ETH Zurich, 8092 Zurich, Switzerland
[a]now at: Biometry and Environmental Systems Analysis, University of Freiburg, 79106 Freiburg i. Br., Germany

*Correspondence to*: Matthias Speich (matthias.speich@wsl.ch)

**Abstract.** Rooting zone water storage capacity $S_r$ is a crucial parameter for modeling hydrology, ecosystem gas exchange and vegetation dynamics. Despite its importance, this parameter is still poorly constrained and subject to high uncertainty. We tested the analytical, optimality-based model of effective rooting depth proposed by Guswa (2008, 2010) with regard to

its applicability for parameterizing $S_r$ in temperate forests. The model assumes that plants dimension their rooting systems to maximize net carbon gain. Results from this model were compared against values obtained by calibrating a local water balance model against latent heat flux and soil moisture observations from 15 eddy covariance sites. Then, the effect of optimality-based $S_r$ estimates on the performance of local water balance predictions was assessed during model validation. The agreement between calibrated and optimality-based $S_r$ varied greatly across climates and forest types. At a majority of

cold and temperate sites, the $S_r$ estimates were similar for both methods, and the water balance model performed equally well when parameterized with calibrated and with optimality-based $S_r$. At spruce-dominated sites, optimality-based $S_r$ were much larger than calibrated values. However, this did not affect the performance of the water-balance model. On the other hand, at the Mediterranean sites considered in this study, optimality-based $S_r$ were consistently much smaller than calibrated values. The same was the case at pine-dominated sites on sandy soils. Accordingly, performance of the water balance model

was much worse at these sites when optimality-based $S_r$ were used. This rooting depth parameterization might be used in dynamic (eco)hydrological models under cold and temperate conditions, either to estimate $S_r$ without calibration or as a model component. This could greatly increase the reliability of transient climate-impact assessment studies. On the other hand, the results from this study do not warrant the application of this model to Mediterranean climates or on very coarse soils. While the cause for these mismatches cannot be determined with certainty, it is possible that trees under these

conditions follow rooting strategies that differ from the carbon budget optimization assumed by the model.

# 1 Introduction

Rooting zone storage capacity $S_r$, expressing the maximum amount of water that can be stored in the soil and accessed by plants, is a crucial variable for the water balance and vegetation dynamics of terrestrial ecosystems. From a hydrological point of view, $S_r$ governs the partitioning of rainfall into transpiration and water yield (Milly, 1994), so that an increase in $S_r$ leads to an increase in long-term transpiration (Federer et al., 2003) and a decrease in long-term runoff (Donohue et al., 2012). Also, as $S_r$ constrains transpiration, it may limit biological productivity (Porporato et al., 2004). Furthermore, $S_r$ is also an important variable controlling water, carbon and energy fluxes at the Earth's surface in climate models (Kleidon and Heimann, 1998; Wang and Dickinson, 2012).

Although its importance has long been recognized, $S_r$ is still a poorly constrained parameter. As $S_r$ is not a directly observable quantity, it is difficult to relate it to field measurements. An often-used useful simplification (Federer et al., 2003; Kleidon and Heimann, 1998) is the definition of $S_r$ (expressed in mm water depth) as the product of the water-holding capacity $\kappa$ [mm mm$^{-1}$] of the soil (i.e. the difference between soil water content at field capacity and at the wilting point) and the effective rooting depth $Z_e$ [mm], defined as the lowest depth in the soil profile where water is still accessible to roots. While $\kappa$ is generally assumed to remain constant, some approaches focus on estimating $Z_e$ to parameterize $S_r$. Given that soil properties and rooting patterns vary at spatial scales much smaller than typical spatial discretization units in hydrological and ecosystem models (such as a catchment, grid cell or forest stand), $\kappa$ and $Z_e$ are usually taken as spatial averages. For this reason, point-scale observations of rooting depth cannot be assumed to be representative for a typical modeling unit (Wang-Erlandsson et al., 2016).

In many model applications, $S_r$ is parameterized with a look-up table approach, attributing the same parameter value to all catchments or cells belonging to the same land-cover class and/or soil type. This approach has the disadvantage of neglecting the variability of rooting properties within one vegetation type. Alternatively, $S_r$ is treated as a tuneable parameter and estimated through calibration, at the expense of interpretability. In addition to those drawbacks, these two approaches treat $S_r$ as a time-invariant parameter. However, rooting properties have been shown to adapt to edaphic and climatic conditions (Gentine et al., 2012), and the inclusion of a dynamic $S_r$ in models has the potential to increase the reliability of projections under a changing climate (Savenije and Hrachowitz, 2017). Several approaches have recently been developed to include the dependence of $S_r$ on environmental conditions. The mass balance approach (de Boer-Euser et al., 2016; Gao et al., 2014) assumes that plants develop their rooting systems so that they can withstand a drought of a certain return period. The storage requirement is estimated based on annual maximal soil moisture deficits over a period of several years, in analogy to engineering calculations used to estimate optimal reservoir size. This approach has been used to generate a global dataset of $S_r$ (Wang-Erlandsson et al., 2016) and to calculate a time-varying $S_r$ for a dynamic hydrological model (Nijzink et al., 2016).

Another way to consider the adaptation of vegetation properties is the use of an optimality assumption, i.e. the assumption that vegetation organizes itself in a way that maximizes biological fitness. Eagleson (1982) first introduced optimality

principles to ecohydrology, showing their potential in the reduction of model parameterization requirements. Several objective functions have been proposed, such as the minimization of water stress (Eagleson, 1982) or the maximization of net primary productivity (Kleidon and Heimann, 1998). Schymanski et al. (2009) argue that the maximization of net carbon profit –the difference between the amount of carbon assimilated through photosynthesis and the amount used for respiration-

is a more appropriate objective function, as the carbon not used for growth and maintenance can be invested into seeds, defense compounds or symbiotic relationships, which all contribute to increase an individual's fitness. Furthermore, this approach offers a solution to the trade-off between the sometimes conflicting objectives of stress minimization and productivity maximization (Schymanski et al., 2009).

A number of optimality-based approaches have been proposed to estimate $Z_e$ or other rooting properties, such as the shape of

the root profile (Collins and Bras, 2007; Guswa, 2008; Kleidon and Heimann, 1998; Schymanski et al., 2008). The approach of Guswa (2008) has recently been used by Yang et al. (2016) to calculate $Z_e$ on a global grid. This model (see Sect. 2.1) calculates the optimal rooting depth as the level where the marginal carbon costs of deeper roots starts to outweigh the marginal benefit. Its optimization target is thus similar to the net carbon profit. The model requires an estimation of vegetation properties, as well as long-term climate characteristics. Estimates of $Z_e$ obtained with this approach were used in

a hydrological model (Donohue et al., 2012), leading to a higher performance than other parameterizations (Yang et al., 2016). The original version of the model, which has been used in these studies, assumes an intensive water uptake strategy, typical for short-lived vegetation. Guswa (2010) proposed an alternative version of the model, with a water uptake strategy corresponding to the more conservative behavior of trees. While the behavior of both models is similar across most climatic conditions, the rooting depths obtained with the 2010 version are substantially larger than with the 2008 version in energy-

limited systems.

The aim of this paper is to assess the suitability of Guswa's 2008 and 2010 models for implementation in a dynamic hydrological or ecohydrological model. A dynamic $S_r$ parameterization in a hydrological model is suitable if (1) it gives sensible estimates of $S_r$ (or rooting depth) for a given combination of climate, soil and above-ground vegetation, (2) its variations across different climates, soil conditions and vegetation types are physiologically and ecologically justifiable, and

(3) the associated uncertainty remains within reasonable bounds. We therefore ask: (a) How well do the predictions of this model agree with values obtained through calibration? (b) How does using optimality-based $S_r$ affect the performance of a local water balance model? (c) How does the sensitivity of this rooting depth model to its various inputs vary across sites? Can these variations be explained physiologically and ecologically? (d) Given the uncertainty of the inputs to this model, how large is the uncertainty of estimated $S_r$ under different climate/soil/vegetation type combinations?

First, to increase the general applicability of the 2010 model, we provide a numerical method to approximate its results. We present an implementation of the model that calculates the rooting zone storage for both the overstory and understory. Then, we compare estimates of $S_r$ obtained with this parameterization against $S_r$ values obtained by calibrating a local water balance model against observations of latent heat flux and soil water content at 15 eddy covariance sites of the FLUXNET network (Baldocchi et al., 2001). We assess the effect of using optimality-based $S_r$ estimates on the performance of the local

water balance model during validation. We also investigate the differences in $S_r$ estimates obtained with the two versions of Guswa's model, as well as the sensitivity of model estimates to its inputs and parameters. We also explore the sensitivity of the model to its inputs, as well as the propagation of uncertainty from the model's inputs to its $S_r$ estimates.

## 2 Methods

**2.1 Guswa's optimal rooting depth models**

**2.1.1 Model concepts**

The optimal rooting depth model of Guswa (2008) was developed as a framework to study the effect of climate, soil and vegetation properties on rooting depth. Although its original purpose was to provide process insight, it has been used to generate estimates of $Z_e$ in studies at regional (Donohue et al., 2012; Smettem and Callow, 2014) and global (Yang et al.,
2016) scale. The fundamental assumption of the model is that plants develop their rooting systems in a way that maximizes net carbon gain. The model compares the benefits of deeper roots (additional carbon uptake through increased transpiration) with the associated costs (construction and maintenance respiration), and sets the optimal rooting depth at the level where the marginal cost equals the marginal benefit. This is expressed as:

$$\frac{\gamma_r \times D_r}{L_r} = w_{ph} \times f_{seas} \times \frac{d\langle T \rangle}{dZ_e},$$

15 (1)

where $\gamma_r$ is root respiration rate [mg C g$^{-1}$ roots day$^{-1}$], $D_r$ root length density [cm roots cm$^{-3}$ soil], $L_r$ specific root length [cm roots g$^{-1}$ roots], $w_{ph}$ photosynthetic water use efficiency [g C cm$^{-3}$ H$_2$O], $f_{seas}$ growing season length [fraction of a year] and $\langle T \rangle$ mean daily transpiration[mm day$^{-1}$] during the growing season (a list of all symbols used in this paper is given in Table A1). The left hand side of Eq. 1 represents the marginal cost of an increase in rooting depth, and the right hand side
represents the associated benefit. The last term in Eq. 1 requires the definition of a function relating average transpiration to rooting depth. Guswa (2008) uses the stochastic model of Milly (1993). This model treats precipitation as a Poisson process, characterized by frequency $\lambda$ [events per day] and average depth $\alpha$ [mm per event]. Such a formulation has been used in many ecohydrological studies at the daily timescale (Porporato et al., 2004; Rodriguez-Iturbe et al., 1999). Transpiration is then expressed as

$\langle T \rangle = \alpha \lambda \frac{exp[\kappa Z_e/\alpha(1-W)]-1}{exp[\kappa Z_e/\alpha(1-W)]-W}.$ (2)

where $\kappa$ is the water holding capacity of the soil [mm water depth/mm soil depth] and W the ratio of effective precipitation $P_{eff}$ and potential transpiration $T_{pot}$. $P_{eff}$ is mean daily precipitation available for transpiration (i.e. minus interception and soil evaporation) and $T_{pot}$ is a hypothetical daily transpiration assuming no soil moisture stress [both in mm day$^{-1}$]. Substituting Eq. 2 into Eq. 1 and solving for $Z_e$ gives

$Z_e = \frac{\alpha}{\kappa(1-W)} ln(X),$ (3)

where X is defined as:

$$X = \begin{cases} W\left[1 + \frac{\kappa}{\alpha}\frac{(1-W)^2}{2A} - \sqrt{\frac{\kappa}{\alpha}\frac{(1-W)^2}{A} + \left(\frac{\kappa}{\alpha}\frac{(1-W)^2}{2A}\right)^2}\right] & if\ W > 1 \\ W\left[1 + \frac{\kappa}{\alpha}\frac{(1-W)^2}{2A} + \sqrt{\frac{\kappa}{\alpha}\frac{(1-W)^2}{A} + \left(\frac{\kappa}{\alpha}\frac{(1-W)^2}{2A}\right)^2}\right] & if\ W < 1 \end{cases} \text{, and} \tag{4}$$

$$A = \frac{\gamma_r \times D_r}{L_r \times w_{ph} \times T_{pot} \times f_{seas}}. \tag{5}$$

For a full derivation of Eqs. 3 to 5, we refer to Guswa (2008).

The transpiration model of Milly (1993) (Eq. 2) assumes that the vegetation transpires at potential rate as long as there is available water in the soil, and that transpiration ceases when the soil moisture reservoir is depleted. This reflects a water uptake strategy typical for many grasses, which tend to maximize carbon assimilation and seed production when water is available, and enter a dormant state or die in drier periods. As long-lived organisms, trees generally have a more conservative water uptake strategy (Chaves, 2002). To examine the effect of water uptake strategy on modeled rooting depth,
Guswa (2010) proposed an alternative version of the optimal rooting depth model, where Eq. 2 is replaced with another function, formulated by Porporato et al. (2004):

$$\langle T \rangle = T_{pot}W - \frac{\exp{(-Z_n)}Z_n^{WZ_n-1}}{\gamma(WZ_n, Z_n)}, \tag{6}$$

where $\gamma(\cdot, \cdot)$ is the lower incomplete gamma function (Weisstein, n.d.), and $Z_n$ is rooting depth expressed as the number of average precipitation events that can be stored within the rooting zone. $Z_n$ is related to the effective rooting depth $Z_e$ through
the following relationship:

$$Z_n = \frac{\kappa Z_e}{\alpha}. \tag{7}$$

This model assumes a linear decrease of transpiration with decreasing soil water content, and reflects the more conservative water uptake strategy of trees.

In both studies, $Z_e$ is at its maximum when water supply and demand are approximately equal. In energy-limited
environments, $Z_e$ is more sensitive to changes in rainfall frequency rather than average depth, while the opposite is true under water-limited conditions. The more conservative water-use strategy consistently leads to deeper roots when all parameters are equal, especially under energy-limited conditions. In the rest of this paper, the two versions of Guswa's optimal rooting depth model will be referred to as G08 and G10. The two implementations presented here calculate a storage volume for both the overstory and understory. In both bases, G08 is used for the understory. One version uses G08 for the
overstory, and the other version uses G10. These two implementations are referred to as G-For08 and G-For10, respectively. Statements that apply to both implementations will use the term G-For.

**2.1.2 Implementation**

In the original model description, soil evaporation is treated as a loss and subtracted from the water and energy balances. In the implementations presented here, instead, it is assumed that there is no soil evaporation, but that sub-canopy evaporation

comes from understory transpiration. As a first approximation, the competition aspect is neglected here, and stand-scale $S_r$ is defined as the sum of storage volumes for the trees and for the understory. For temperate forests, one can generally assume that the forest floor is covered with a layer of shrubs or non-woody plants, and that bare soil evaporation is negligible. The storage volume for the understory can then in turn be estimated assuming that its rooting system is optimized, as constrained by the amount of energy reaching the forest floor. Donohue et al. (2012) use a similar approach, by first calculating an optimal rooting depth for both trees and grasses, and providing a grid-cell average by weighting these two values with the respective fractional cover. Here, the values for the overstory and understory are weighted by the fraction of light that is intercepted by the canopy and that reaches the ground, respectively. The light partitioning is calculated using Beer's law. Figure 1 shows the structure of a sample forest stand, and the simplifying assumptions made here. Despite their spatial heterogeneity, above- and belowground vegetation and site characteristics are assigned a single value. Partitioning of incoming water and available energy is governed by the leaf area index (LAI) of the overstory.

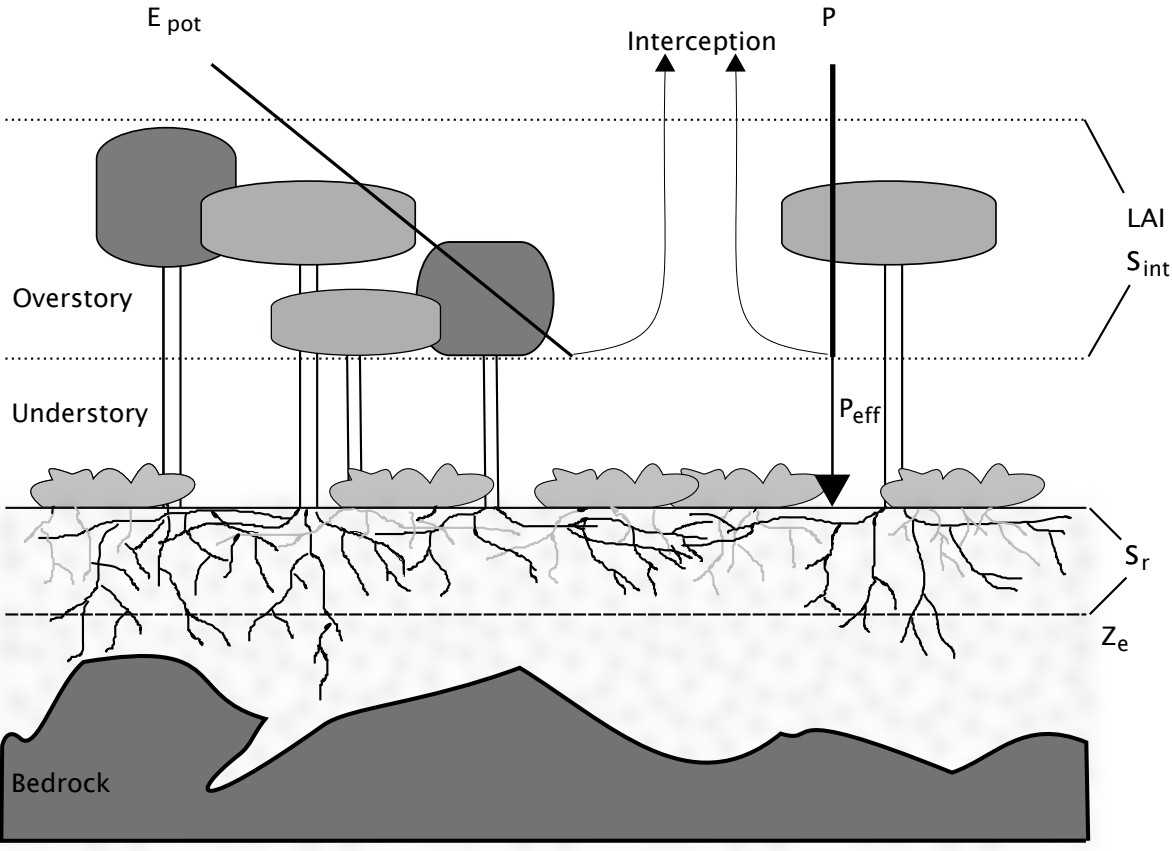

**Figure 1: Schematic representation of a forest stand, together with the simplifications used in this study. The stand is heterogeneous in terms of overstory and understory density, as well as soil depth. In the model, both aboveground and belowground properties are integrated to stand-level variables. The crowns of the overstory trees form a canopy described by the variables leaf area index (LAI) and interception storage capacity ($S_{int}$). LAI determines the partitioning of available energy $E_{pot}$ between potential transpiration of the overstory and understory. Incoming precipitation is divided between effective precipitation**

**reaching the ground $P_{eff}$, and interception. No distinction is made between understory transpiration, understory interception evaporation and soil evaporation. Below ground, rooting depth is expressed as a stand-scale average ($Z_e$). Rooting zone water storage capacity $S_r$ is the product of $Z_e$ and soil water holding capacity, assumed to be constant over the whole stand, despite its high horizontal and vertical heterogeneity in reality.**

Available energy is represented by mean daily Penman (1948) potential evaporation ($E_{pot}$). The effective amount available to the vegetation (including both understory and overstory) is set to $0.75*E_{pot}$. The factor 0.75 accounts for the energy used for interception evaporation and for stomatal and aerodynamic resistances, and was set based on the meta-analysis of Granier et al. (1999).

In the G-For08 implementation, the G08 model (Eqs. 2 to 5) is used to calculate the storage capacity for both the understory and overstory:

$$S_r(GFor08) = G08(climate, \kappa, V_{tree}) \times (1 - e^{-k_l LAI}) + G08(climate, \kappa, V_{grass}) \times e^{-k_l LAI}, \tag{8}$$

where $k_l$ is the canopy light extinction coefficient (taken as 0.5), LAI is overstory leaf area index during the growing season, and $V_{tree}$ and $V_{grass}$ are the vegetation parameter sets for trees and grass, respectively, given in Table 1 (see also Sect. 2.1.4

below). In the G-For10 implementation, storage capacity for the overstory is calculated with the G10 model:

$$S_r(GFor10) = G10(climate, \kappa, V_{tree}) \times (1 - e^{-k_l LAI}) + G08(climate, \kappa, V_{grass}) \times e^{-k_l LAI}, \tag{9}$$

As differentiating and rearranging the model of Porporato et al. (2004) (Eq. 6) leads to rather cumbersome expressions, an approximation was used here for the G10 model. It follows from Eq. 1 that the optimal rooting depth is the value of $Z_e$ for which $dT/dZ_e$ equals the ratio $\gamma_r D_r / L_r w_{ph} f_{seas}$. Therefore, the optimal $Z_e$ is found by applying Eq. 6 to increasing values

of $Z_e$, until the difference to the previous iteration is less than or equal to that ratio.

**Table 1: Values of the vegetation parameters needed for the optimal rooting depth model, based on Donohue et al. (2012).**

| Parameter | Trees | Grass |
| --- | --- | --- |
| $w_{ph}$ [mmol $CO_2$ $cm^{-3}$ water ] | 0.33 | 0.22 |
| $\gamma_{r,20}$ [mmol $CO_2$ $g^{-1}$ roots $day^{-1}$] | 0.5 | 0.5 |
| $L_r$ [cm roots $g^{-1}$ roots] | 1500 | 1500 |
| $D_r$ [cm roots $cm^{-3}$ soil] | 0.1 | 0.1 |
| $f_{seas}$ [fraction of a year] | (See Sect. 2.1.3) | 0.7 |

**2.1.3 Parameterization**

In the present study, the climate parameters are derived from daily averaged measurements of air temperature, precipitation,
vapor pressure deficit (VPD), global radiation and wind speed at 15 FLUXNET sites (see Sect. 2.2.1 below). To define the start of the growing season for trees, the species-specific spring phenology model developed and parameterized by Kramer

(1996) was applied at each site, with the parameters corresponding to the dominant species. Following Zierl (2001), the onset of leaf senescence in autumn was set to the first time the four-day mean temperature drops below 5°C. The end of the growing season is set to 14 days after the onset of leaf senescence. For *Pinus pinaster*, for which no species-specific parameters were available, the growing season was assumed to last from April to October. For the understory, the growing

season duration $f_{seas}$ was set to 0.7 (Table 1). Potential evaporation was calculated using the Penman (1948) equation and averaged to mean daily values over the growing season. To calculate precipitation frequency $\lambda$ and average depth $\alpha$, a precipitation event was defined as a period of one or more consecutive days with precipitation greater than 0.5 mm/day. Effective precipitation $P_{eff}$ was estimated as follows (Guswa, 2008):

$$P_{eff} = \alpha\lambda \times exp(-S_{int}/\alpha), \tag{10}$$

where $S_{int}$ is the canopy interception storage capacity [mm]. This value was estimated from LAI using the relationship proposed by Menzel (1997) and Vegas Galdos et al. (2012):

$$S_{int} = k_{int} \times log_{10}(1 + LAI), \tag{11}$$

where $k_{int}$ is an empirical parameter, set to 1.6 for broadleaved forests, 1.8 for mixed forests and 2 for coniferous forests (Vegas–Galdos et al., 2012).

The vegetation parameters were taken from Donohue et al. (2012), who compiled them from values found in the literature. The parameter values for trees and grass are shown in Table 1. Root respiration rate is parameterized as a function of temperature, following Yang et al. (2016):

$$\gamma_r = \gamma_{r,20}Q_{10}^{\left(\frac{T-20}{10}\right)}, \tag{12}$$

where T is the mean soil temperature over the growing season, and $Q_{10}$ is a coefficient indicating the effect of a 10 K rise in

temperature. In the absence of soil temperature measurements, air temperature can be taken as a proxy (Yang et al., 2016). Based on the experimental findings of Keller (1967), $Q_{10}$ was set to 2.

## 2.2 $S_r$ estimated through model calibration

As mentioned above, $S_r$ and $Z_e$ are model parameters that cannot be directly measured in the field. Due to the high spatial heterogeneity of rooting depth and soil properties, field measurements of rooting depth are not necessarily indicative of the

average conditions in a forest stand. An alternative to measurements is the estimation of parameter values through model calibration (Gao et al., 2014). In this study, $S_r$ was estimated at 15 eddy covariance sites from the FLUXNET network (Baldocchi et al., 2001) by calibrating the local water balance model FORHYTM (Forest Hydrology Toy Model; Speich et al. (2018); see https://github.com/mspeich/forhytm). Modeled total evaporation ($E_{tot}$, defined as the sum of canopy transpiration, soil and understory evaporation and interception evaporation) and relative extractable water (REW; see below)

were compared against measurements at half-hourly time steps.

**2.2.1 FLUXNET site selection**

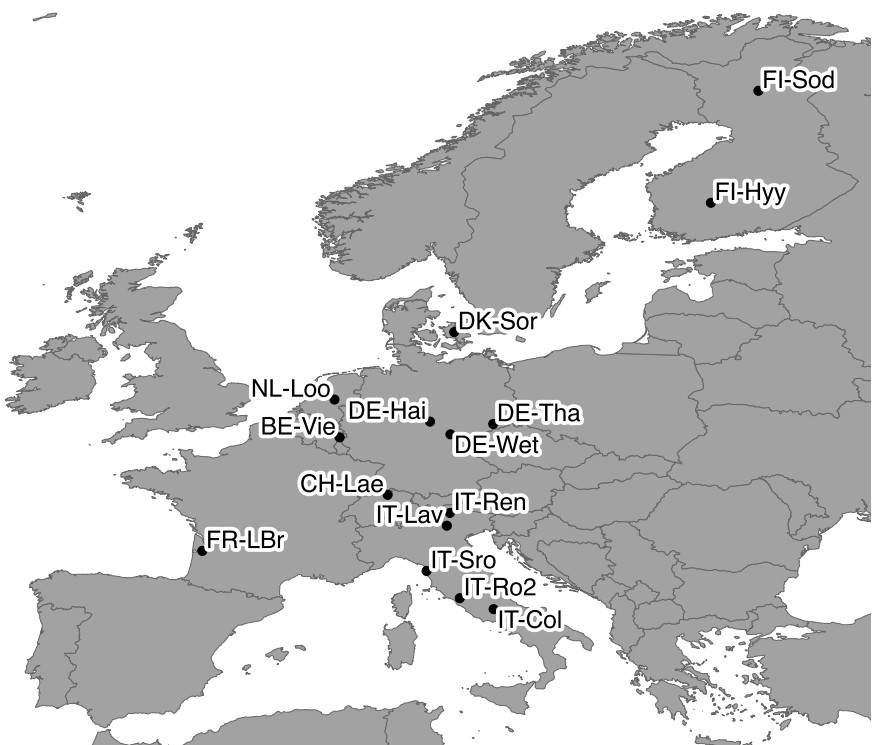

**Figure 2: Map of the 15 FLUXNET sites used in this study. Base map elements from Natural Earth.**

Table 2 gives an overview of the FLUXNET sites used in this analysis, and their location is shown on Fig. 2. The conditions for site selection were the following: (1) at least four years of continuous latent heat flux measurements in the FLUXNET-2015 (Tier 1) or La Thuile (fair use) datasets; (2) belonging to a forested IGBP land cover class (either Evergreen Needleleaf Forest (ENF), Evergreen Broadleaf Forest (ENF), Deciduous Broadleaf Forest (DBF), Deciduous Needleleaf Forest (DNF) or Mixed Forest (MF)); (3) temperate or cold climate (group C or D in the Köppen-Geiger (Köppen, 2011) classification);

(4) no a priori indications (e.g. in the site description) of a shallow water table or irrigation; (5) availability of soil water content (SWC) measurements at a depth that can be taken as representative for the average conditions in the rooting zone. The last criterion greatly limits the number of sites retained in this analysis, as for many sites, the soil water measurements are representative for the near-surface conditions only. It is however necessary to exclude these sites, as the absolute values and dynamics of soil moisture in the uppermost layers can differ greatly from the conditions at greater depths (Miller et al.,

2007). For each site, the suitability of SWC measurements was determined through a subjective assessment of the SWC curves. The soil moisture content at field capacity $\theta_{FC}$ was estimated by eye as the level where SWC stabilizes after a refilling event, and the soil moisture content at the wilting point $\theta_{WP}$ was assumed to correspond to the lowest SWC

measured over the whole period. The corresponding soil water holding capacity $\kappa$, i.e. the difference between $\theta_{FC}$ and $\theta_{WP}$, is reported in Table 2.

**Table 2: Overview of the FLUXNET sites used in this study. Where a model validation was performed, the validation period is given in brackets. LAI refers to the value at full foliage.**

| Site | Years used | Lat/Lon | m asl | Dominant species | LAI | $f_c$ | $\kappa$ [mm/mm] | Reference |
|------|-----------|---------|-------|------------------|-----|-------|------------------|-----------|
| Vielsalm (BE-Vie) | 1997-2008 (2010-2012) | 50.3, 6 | 491 | Fagus sylvatica | 4.5 | 0.9 | 0.11 | Aubinet et al. (2001) |
| Lägeren (CH-Lae) | 2005-2010 (2011-2013) | 47.45, 8.4 | 689 | Fagus sylvatica | 5.5 | 0.9 | 0.12 | Etzold et al. (2011) |
| Hainich (DE-Hai) | 2004-2009 (2000-2003) | 51.1, 10.5 | 430 | Fagus sylvatica | 5 | 0.9 | 0.28 | Anthoni et al. (2004) |
| Tharandt (DE-Tha) | 1997-2003 (2004-2008) | 51, 13.6 | 320 | Picea abies | 7.2 | 0.9 | 0.15 | Grünwald and Bernhofer (2007) |
| Wetzstein (DE-Wet) | 2003-2006 | 50.5, 11.5 | 703 | Picea abies | 4 | 0.9 | 0.19 | Anthoni et al. (2004) |
| Sorø (DK-Sor) | 2008-2013 (2005-2006) | 55.5, 11.6 | 40 | Fagus sylvatica | 5 | 0.9 | 0.19 | Wang et al. (2005) |
| Hyytiälä (FI-Hyy) | 2003-2007 (2008-2013) | 61.8, 24.3 | 181 | Pinus sylvestris | 3.3 | 0.45 | 0.3 | Lindroth et al. (2008) |
| Sodankylä (FI-Sod) | 2001-2006 (2007-2010) | 67.4, 26.6 | 188 | Pinus sylvestris | 1.7 | 0.45 | 0.06 | Lindroth et al. (2008) |
| Le Bray (FR-LBr) | 2003-2008 (2010-2012) | 44.7, -0.8 | 62 | Pinus pinaster | 2.8 | 0.8 | 0.11 | Loustau et al. (2005) |
| Collelongo (IT-Col) | 2007-2012 (1997-2001) | 41.8, 13.6 | 1560 | Fagus sylvatica | 4.5 | 0.8 | 0.17 | Valentini et al. (1996) |
| Lavarone (IT-Lav) | 2004-2010 (2011-2014) | 46, 11.3 | 1305 | Abies alba | 9.6 | 0.9 | 0.25 | Cescatti and Marcolla (2004) |
| Renon (IT-Ren) | 2005-2009 (2002-2003) | 46.6, 11.4 | 1794 | Picea abies | 5.5 | 0.9 | 0.23 | Cescatti and Marcolla (2004) |
| Roccarespam- | 2003-2008 | 42.4, | 160 | Quercus | 4.5 | 0.9 | 0.14 | Chiti et al. (2010) |

| | | | | | | | | |
|---|---|---|---|---|---|---|---|---|
| pani 2 (IT-Ro2) | (2010-2012) | 11.9 | | cerris | | | | |
| San Rossore (IT-SRo) | 2000-2006 (2007-2009) | 43.7, 10.3 | 6 | Pinus pinaster | 2.8 | 0.5 | 0.06 | Chiti et al. (2010) |
| Loobos (NL-Loo) | 1997-2007 (2008-2013) | 52.2, 5.7 | 25 | Pinus sylvestris | 3 | 0.8 | 0.05 | Kramer et al. (2002) |

## 2.2.2 Model calibration, parameter estimation and validation

The local water balance model FORHYTM was calibrated at each site to obtain estimates of $S_r$. As shown on Fig. B1, the model contains two state variables, the interception and plant-available soil moisture reservoirs. The former is filled by incoming precipitation and emptied by interception evaporation. The latter is filled by effective precipitation (after subtracting the intercepted fraction) and depleted by canopy transpiration and soil/understory evaporation. A full description of the model is given in Speich et al. (2018), and a summary is given in Appendix B. Based on the screening analysis of Speich et al. (2018), seven parameters, including $S_r$, were selected for calibration. These parameters are listed in Table B1. Modeled total evaporation ($E_{tot}$) and soil moisture were compared against measurements of latent heat flux and soil water content (SWC). SWC measurements were converted to relative extractable water (Granier et al., 2007) as follows:

$$REW = min\left(1, \frac{\theta - \theta_{WP}}{\theta_{FC} - \theta_{WP}}\right). \tag{13}$$

For both outputs, the goodness-of-fit measure is the Kling-Gupta efficiency $KGE$ (Gupta et al., 2009) with the slight modification proposed by Kling et al. (2012). $KGE$ is defined as:

$$KGE = 1 - \sqrt{(r-1)^2 + (\beta-1)^2 + (\gamma-1)^2}, \tag{14}$$

where $r$ is the Pearson correlation coefficient between the simulated and observed values, $\beta$ the bias ratio (ratio of the means of the simulated and observed values), and $\gamma$ the variability ratio (ratio of the coefficients of variation of the simulated and observed values). The final criterion used to determine the goodness-of-fit, $KGE_{AVG}$, is the average of the $KGE$ values obtained for TE and REW. Only the time steps that are part of the growing season (given by the phenology model) were considered. Furthermore, time steps where the quality control flag indicated unreliable observations were excluded.

An overview of the calibration and validation runs is given in Table 3. During calibration, FORHYTM was run at each site with 1000 different combinations of parameter values, sampled from the parameter space given in Table B1 using the Latin Hypercube Sampling procedure of Beachkofski and Grandhi (2002). At each site, the parameter sets with $KGE_{AVG}$ scores equal or greater than the 95[th] percentile ($P_{95}$) were retained for model validation (Validation_Calibrated in Table 3). To assess the suitability of G-For08 and G-For10 estimates of $S_r$ for water balance modeling, two additional sets of runs were performed over the validation period (Validation_G-For08 and Validation_G-For10). In these runs, the parameter sets were the same as for Validation_Calibrated, but $S_r$ was replaced with the value estimated with G-For08 and G-For10,

respectively. Table 2 lists the calibration and validation periods at each site. Where soil water content measurements were available for the calibration period only, validation was only performed against $E_{tot}$. Furthermore, as only five years of measurements are available for Wetzstein, no validation was undertaken for that site.

**Table 3: Overview of calibration and validation runs of the FORHYTM model.**

| Set of model runs | Parameter sets | # runs per site | Period |
|---|---|---|---|
| Calibration | Latin Hypercube Sampling | 1000 | Calibration period (Table 2) |
| Validation_Calibrated | Parameter sets from Calibration runs where $KGE_{avg} \geq P_{95}$ (site-specific) | $\geq 50$ | Validation period (Table 2) |
| Validation_G-For08 | Same as for Validation_Calibrated, but with $S_r$ estimated with G-For08 estimate | $\geq 50$ | Validation period (Table 2) |
| Validation_G-For10 | Same as for Validation_Calibrated, but with $S_r$ estimated with G-For10 estimate | $\geq 50$ | Validation period (Table 2) |

## 2.3 Uncertainty and sensitivity analyses

The use of calibration to estimate $S_r$ presupposes that the dynamic model is highly sensitive to this parameter. A sensitivity analysis of FORHYTM (Speich et al., 2018) revealed that $S_r$ is one of the most influential parameters for long-term water balance. To assess whether this is also the case for intra-annual dynamics of evaporation and soil moisture, a new sensitivity 
analysis was conducted here, examining the effect of all calibration parameters on $KGE_{AVG}$.

For the rooting depth models, on the other hand, parameter values are either fixed (the plant physiological parameters), estimated from site characteristics (e.g. LAI or soil water holding capacity $\kappa$) or calculated from micrometeorological measurements. Each of these inputs is subject to uncertainty. For example, the plant physiological parameters compiled by Donohue et al. (2012) are based on ranges reported in the literature, and might vary with species, size and location. Site 
parameters, especially $\kappa$, represent quantities with a high spatial heterogeneity, so that the values used here might not be representative of the entire footprint. The climate parameters are influenced by the micrometeorological measurement uncertainty. Furthermore, the meteorological record used here only spans a couple of years at each site and is not necessarily representative of the long-term climatic conditions that have influenced rooting depth. It is therefore necessary to examine how this uncertainty propagates to model outputs. Also, for future uses of the model it is useful to know which parameters
contribute most to the uncertainty of $S_r$. Therefore, a sensitivity and uncertainty analysis was also conducted for G-For10.

The approach chosen for both sensitivity analyses (FORHYTM and the rooting depth model G-For10) was similar: the models were first run multiple times with varying parameter values. Then, a statistical meta-model was fitted, with the parameters as predictors and the target variable ($KGE_{AVG}$ in the case of FORHYTM, and $S_r$ for G-For10) as the dependent variable. The selected meta-modeling procedure is Random Forest (Breiman, 2001), a bootstrapped and randomized

ensemble of regression trees. The Random Forest procedure possesses several useful properties for this application: it can handle nonlinear effects and parameter interactions, requires a relatively small number of simulations and provides a variable importance ranking (Harper et al., 2011). The variable importance measure used here is the Mean Decrease in Accuracy (Liaw and Wiener, 2002), which expresses the increase in model prediction error when the values of a predictor are permutated (i.e. converted to random noise). Due to the non-deterministic nature of Random Forest, the variable importance measures vary with each application. The ranking of parameters, however, is generally more stable. Two parameters of Random Forest itself affect the stability of variable importance rankings: the number of regression trees $ntree$ and $mtry$, the number of variables used at each split (Genuer et al., 2010). The number of trees should be high enough for the model to converge, and increasing $mtry$ leads to greater differences between the importance measures of the different parameters, thus increasing the stability of rankings. In the analyses presented here, the stability of rankings was assessed by comparing the outcome of several Random Forest runs, and $ntree$ and $mtry$ were adapted if necessary.

For FORHYTM, the sensitivity analysis was performed directly on the calibration runs. The number of regression trees was set to 5000, and $mtry$ to its default value of 2. For G-For10, 2000 parameter sets were generated, with perturbations of all parameters by up to 20%. The parameters include the plant physiological parameters for trees and grass, climate statistics and site characteristics. In addition, the start and end of the growing season were also shifted back or forward by up to 10 days (which, in turn, also affects the climate statistics calculated over the growing season). As the plant physiological parameters are multiplied with each other only and do not interact with other variables individually, they are condensed into two variables, $PP_o$ for the overstory and $PP_u$ for the understory. The overstory parameter is defined as:

$$PP_o = \frac{\gamma_{r,20}D_r}{L_r w_{ph}}. \tag{15}$$

Using the parameter values for trees listed in Table 1, $PP_o$ has a standard value of 0.0001. A higher value corresponds to higher costs of additional roots. For the understory, the definition of $PP_u$ is slightly different, as the growing season length is also prescribed:

$$PP_u = \frac{\gamma_{r,20}D_r}{L_r w_{ph} f_{seas}}. \tag{16}$$

Using the parameter values for grass (Table 1), $PP_u$ has a standard value of 0.000216. For both understory and understory, this parameter does not yet include the effect of temperature, which affects root respiration rate as per Eq. 12.

All parameters used in the sensitivity analysis of G-For10 are marked with an asterisk in Table A1. Sampling was again done with the Latin Hypercube method, and a uniform distribution was assumed for each parameter within their ± 20% range. Here, $ntree$ was set to 5000 and $mtry$ to 10. Unlike for FORHYTM, the question here is not how variations in absolute parameter values affect $S_r$ estimates, but how the uncertainty of these parameters propagates to $S_r$ estimates. Therefore, the predictors are not the absolute parameter values, but a normalized variable indicating their perturbation, with 0 corresponding to a perturbation of -20%, 0.5 to no perturbation, and 1 to a perturbation of +20%.

## 3 Results

### 3.1 Calibrated $S_r$ estimates

Figure 3 shows the $KGE_{AVG}$ scores obtained during calibration at each site, plotted against the $S_r$ parameter values. The upper limit of the point cloud shows the highest $KGE_{AVG}$ that was obtained for a given $S_r$. At a majority of sites, $KGE_{AVG}$ is

most sensitive to $S_r$ at the lower end of its range, with a sharp increase of maximum $KGE_{AVG}$ with increasing $S_r$ up to an optimum or plateau. At most sites, $KGE_{AVG}$ is also limited by $S_r$ at the upper end of its range, although the slope is generally less steep. The $KGE_{AVG}$ values that are greater than or equal to the 95[th] percentile at each site are shown in dark blue. These "behavioral" model runs cover a contiguous part of the $S_r$ range. However, this range is rather broad at some sites, and it is also possible to obtain a poor fit even with an optimal $S_r$ value. This suggests that other parameters also substantially affect

$KGE_{AVG}$. Nevertheless, the sensitivity analysis of FORHYTM shows that out of the seven calibration parameters, $S_r$ is the third, second or most important parameter at all sites (Table B2). Also, the fraction of the $S_r$ parameter range covered by behavioral simulations is relatively narrow, compared to the two other important parameters, $r_{s,min}$ and $l_{vpd}$ (Fig. S-1 and S-2, respectively). The median and standard deviation of the $S_r$ corresponding to behavioral simulations at each site are taken as the calibration-based $S_r$ estimates and uncertainty measures, respectively. These values are given in the first column of

Table 4, and shown on the x-axes of Fig. 5.

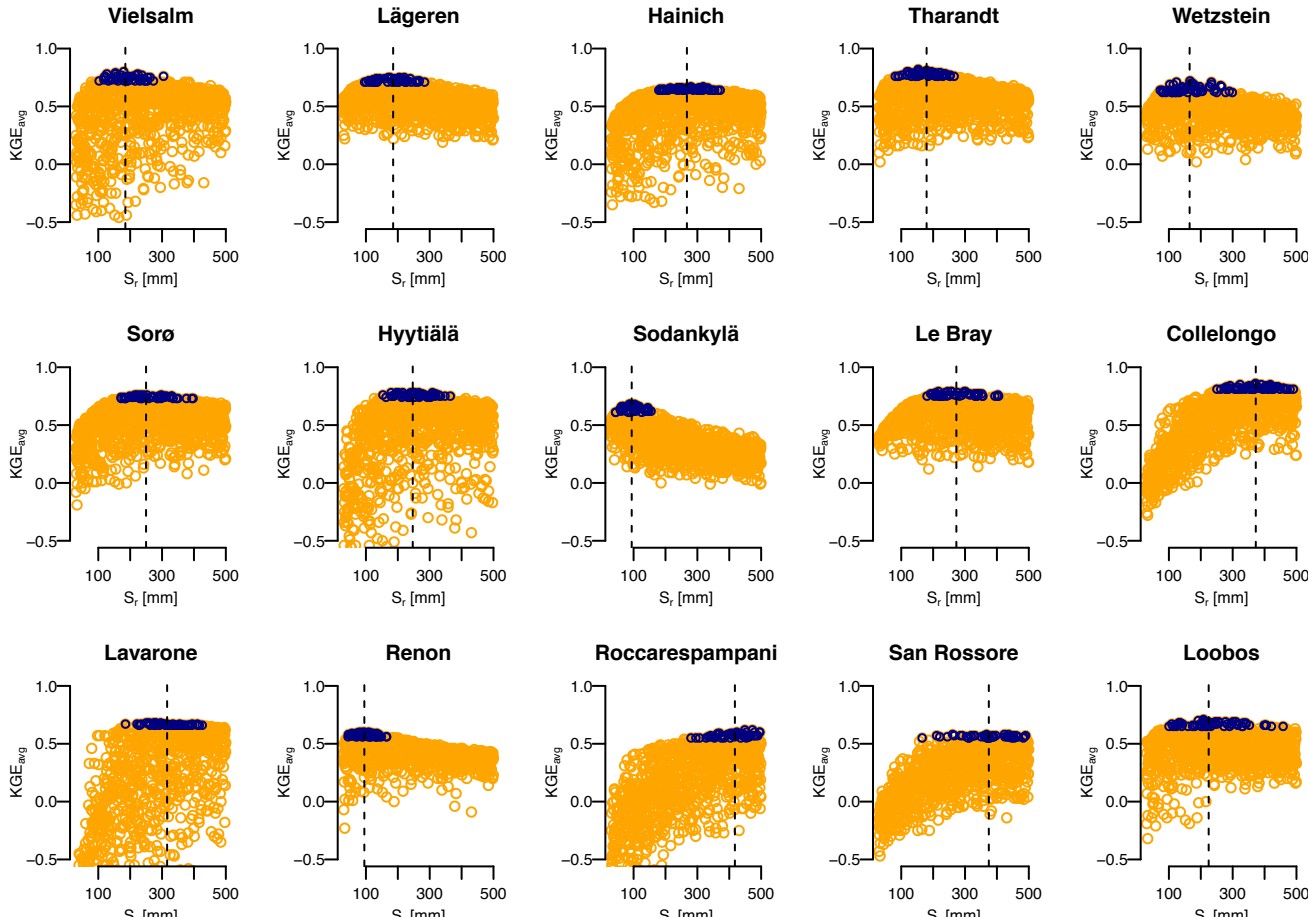

**Figure 3: Relationship of the $KGE_{avg}$ obtained during calibration and the parameter values of $S_r$. The dark blue points have a $KGE_{avg}$ greater or equal to the 95th percentile at this site, and the line shows the median of these values (i.e. the value reported in Table 5). Although $KGE_{avg}$ scores lower than -0.5 occur at some sites, the y-axis has been truncated in these graphs for clarity. At most sites, the $KGE_{avg}$ scores decrease faster with $S_r$ smaller than the optimal range than they do with larger $S_r$.**

**Table 4: Calibrated and modeled $S_r$ obtained at each site. For calibrated values, the $S_r$ value is the median of the parameter values in the simulations with $KGE_{AVG}$ equal or greater than the 95th percentile. The value in parentheses is the standard deviation of these parameter values. For $S_r$ estimates obtained with G-For10, this table shows the values calculated with unperturbed parameters, and the value given in parentheses is the standard deviation of results obtained in the uncertainty analysis.**

| Site | Calibrated $S_r$ [mm] (SD) | G-For08 $S_r$ | G-For10 $S_r$ |
|---|---|---|---|
| Vielsalm | 184 (46) | 128 | 170 (34) |
| Lägeren | 185 (52) | 129 | 187 (39) |
| Hainich | 267 (50) | 255 | 351 (68) |
| Tharandt | 179 (45) | 166 | 230 (46) |

| | | | |
|---|---|---|---|
| Wetzstein | 164 (58) | 191 | 250 (48) |
| Sorø | 249 (59) | 216 | 293 (57) |
| Hyytiälä | 246 (53) | 213 | 283 (56) |
| Sodankylä | 94 (30) | 68 | 70 (14) |
| Le Bray | 272 (61) | 98 | 135 (33) |
| Collelongo | 372 (60) | 141 | 205 (67) |
| Lavarone | 315 (57) | 139 | 297 (67) |
| Renon | 94 (31) | 140 | 241 (57) |
| Roccarespampani | 417 (54) | 105 | 136 (33) |
| San Rossore | 374 (77) | 87 | 100 (20) |
| Loobos | 224 (89) | 84 | 89 (16) |

## 3.2 Climate characteristics of the selected FLUXNET sites

The climate parameters calculated over the calibration period are shown in Table 5. As can be seen on Fig. 4 a), $E_{pot}$ is greater than or approximately equal to $P$ during the growing season at most sites. The high montane sites Lavarone and

5   Renon, as well as the montane site Lägeren, are the only sites where precipitation is clearly greater than $E_{pot}$. Other clusters are formed by the boreal sites, with low $E_{pot}$ and low $P$; the Mediterranean sites and Le Bray, with high $E_{pot}$ and low $P$; and the temperate lowland sites, with low $E_{pot}$ and intermediate $P$. Figure 4 b) shows the distribution of the rainfall properties $\lambda$ (frequency of events) and $\alpha$ (mean depth). Again, the sites located in the Alps and nearby form a cluster, with a high mean precipitation intensity and an intermediate frequency. The Mediterranean sites are characterized by an intermediate $\alpha$ and

10  low $\lambda$, whereas the boreal sites receive precipitation at a high frequency but with a low mean intensity. The temperate lowland sites cover the space between low and intermediate $\alpha$, and between intermediate and high $\lambda$.

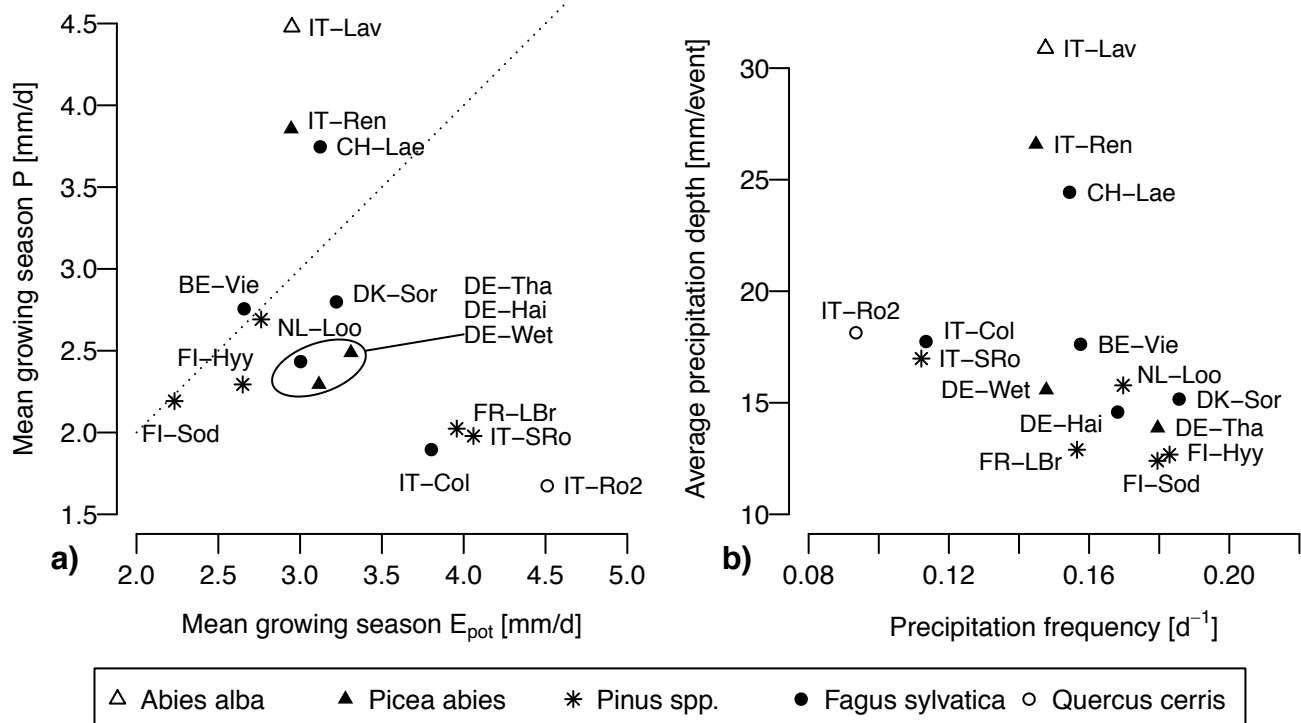

**Figure 4: (a) Position of the 15 selected FLUXNET stations in the $E_{pot}$-$P$ space. The values are daily averages, calculated over the growing seasons of the calibration period. The dotted line is the 1:1 line. Since only the growing season is considered, $E_{pot} > P$ at most sites. (b) Position of the sites in the $\lambda$-$\alpha$ space.**

**Table 5: Climate parameters, calculated as growing-season averages over the calibration period (see text).**

| Site | $P$ | $P_{eff}$ | $\lambda$ | $\alpha$ | $E_{pot}$ | $f_{seas}$ |
|------|-----|-----------|-----------|----------|-----------|------------|
| | [mm/d] | [mm/d] | [1/d] | [mm] | [mm/d] | |
| Vielsalm | 2.76 | 2.55 | 0.16 | 17.62 | 2.66 | 0.47 |
| Lägeren | 3.74 | 3.53 | 0.15 | 24.43 | 3.12 | 0.47 |
| Hainich | 2.43 | 2.23 | 0.17 | 14.58 | 3 | 0.48 |
| Tharandt | 2.48 | 2.18 | 0.18 | 13.87 | 3.31 | 0.45 |
| Wetzstein | 2.29 | 2.1 | 0.15 | 15.58 | 3.12 | 0.4 |
| Sorø | 2.8 | 2.58 | 0.19 | 15.16 | 3.22 | 0.47 |
| Hyytiälä | 2.29 | 2.08 | 0.18 | 12.67 | 2.65 | 0.38 |
| Sodankylä | 2.19 | 2.04 | 0.18 | 12.39 | 2.23 | 0.28 |
| Le Bray | 2.02 | 1.85 | 0.16 | 12.9 | 3.96 | 0.59 |

| Collelongo | 1.89 | 1.77 | 0.11 | 17.74 | 3.8 | 0.42 |
| Lavarone | 4.48 | 4.19 | 0.15 | 30.88 | 2.95 | 0.38 |
| Renon | 3.85 | 3.62 | 0.14 | 26.58 | 2.95 | 0.31 |
| Roccarespampani | 1.67 | 1.57 | 0.09 | 18.13 | 4.51 | 0.52 |
| San Rossore | 1.98 | 1.85 | 0.11 | 16.98 | 4.06 | 0.59 |
| Loobos | 2.69 | 2.49 | 0.17 | 15.78 | 2.76 | 0.48 |

### 3.3 $S_r$ parameterization

The $S_r$ estimates obtained with the G-For08 and G-For10 models are given in the two last columns of Table 4. Figure 5 a) shows the G-For10 estimates plotted against the calibration-based $S_r$. The red horizontal bars indicate the standard deviation
of the calibrated values, as given in Table 4. The horizontal dashes show the median of the $S_r$ estimates obtained at each site with perturbed parameter values, and the blue vertical bars extend to ± one standard deviation from the median, which is represented by the horizontal dash (the results of the sensitivity and uncertainty analyses of the G-For10 model are presented in more detail in Sect. 3.4 below). Some sites show a good agreement between calibrated and modeled $S_r$, such as the boreal pine sites Hyytiälä and Sodankylä, and the beech sites Sorø, Lägeren and Vielsalm. On the other hand, some sites show a
strong disagreement. The optimality-based $S_r$ are much lower than the calibrated value at the Mediterranean sites Roccarespampani and San Rossore (and, to a lesser extent, Collelongo), and at the pine sites Loobos and Le Bray, and are much higher at the spruce sites Tharandt, Wetzstein and Renon. In Fig. 5 b), the G-For estimates are compared against $S_r$ values obtained with Guswa's 2008 model (G-For08). In all cases, G-For10 yielded greater values than G-For08. The differences between both model versions vary greatly across sites, ranging from around 10 mm (Sodankylä, Loobos, San
Rossore) to over 150 mm (Lavarone). In general, the difference is smaller at water-limited sites and at sites with a low water-holding capacity. Also, the greatest differences occur at energy-limited sites with a high water-holding capacity (Lavarone, Hainich, Hyytiälä, Renon).

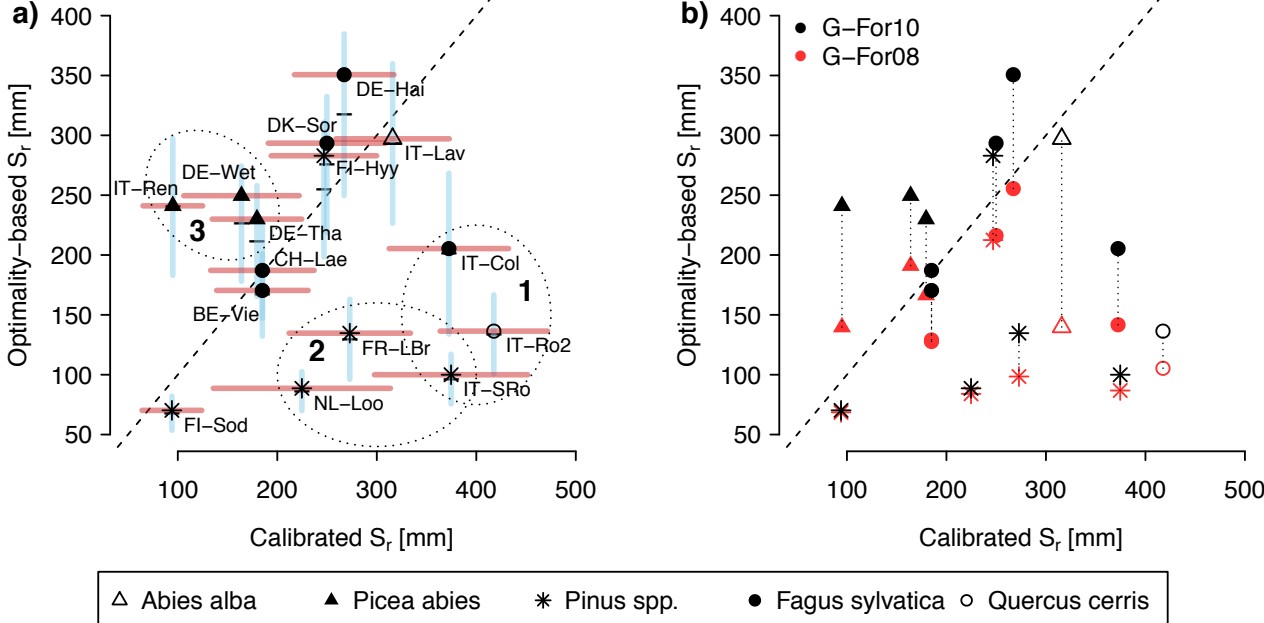

**Figure 5: (a)** Results of the optimality-based $S_r$ estimates obtained with G-For10, plotted against the calibration-based $S_r$. The red horizontal bars show the standard deviation of the calibrated $S_r$ at each site. The point symbols show the $S_r$ estimates obtained with standard parameterization, whereas the horizontal dashes show the median of estimates obtained with perturbed parameter values. The blue vertical bars extend to ± one standard deviation from the median. The ellipses show the cases with strong mismatches between G-For10 and calibrated $S_r$, discussed in Sect. 4.2: Mediterranean climates (1), pine sites on sandy soils (2), and spruce sites along an elevational gradient (3). The dashed line is the 1:1 line. **(b)** Comparison between $S_r$ estimates obtained with G-For10 (same values as on a)), and $S_r$ values obtained using Guswa's 2008 model (G-For08). At all sites, G-For08 estimates are lower than G-For10 estimates. The effect of water uptake strategy varies greatly across sites, with the largest differences occurring at energy-limited sites with a high water-holding capacity (e.g. Lavarone, Hainich).

### 3.4 Parameter sensitivity of G-For10 and uncertainty of $S_r$ estimates

The standard deviation of the $S_r$ obtained with perturbed parameters, a measure of uncertainty of the G-For10 estimates, is given in Table 4 and represented as the blue vertical bars on Fig. 5 a). The standard deviations range between 16 and 68, and are broadly proportional to the $S_r$ value. Figure S16 shows a probability density plot of the $S_r$ values obtained at each station during this uncertainty analysis. It can be seen that the ranges extend quite far to the right at many sites, i.e. some very large values occur at the long tails (up to the double of the median).

Figure 6 shows the contribution of perturbing each parameter to variations in $S_r$. The Random Forest models explained over 80% of the variation in predicted $S_r$ at all sites. At a majority of cold and temperate sites, the main source of variation in $S_r$ are perturbations of potential evaporation $E_{pot}$ and of the physiological parameters of the overstory, represented by the summary parameter $PP_o$ (Eq. 15). At the maritime and/or Mediterranean sites Le Bray, San Rossore and Roccarespampani, perturbations of these parameters are somewhat less important, whereas variations of mean precipitation intensity $\alpha$ have a higher rank. The temperature coefficient $Q_{10}$ is of intermediate importance, except at the warmer sites, where it is less

influential. Soil water holding capacity $\kappa$ is of high or intermediate importance at all sites. Variations in the physiological parameters for grass have very little effect on the $S_r$ estimates. Varying the start and end of the growing season by ± ten days is generally of little importance, except at the colder sites (spring) and under Mediterranean conditions (autumn).

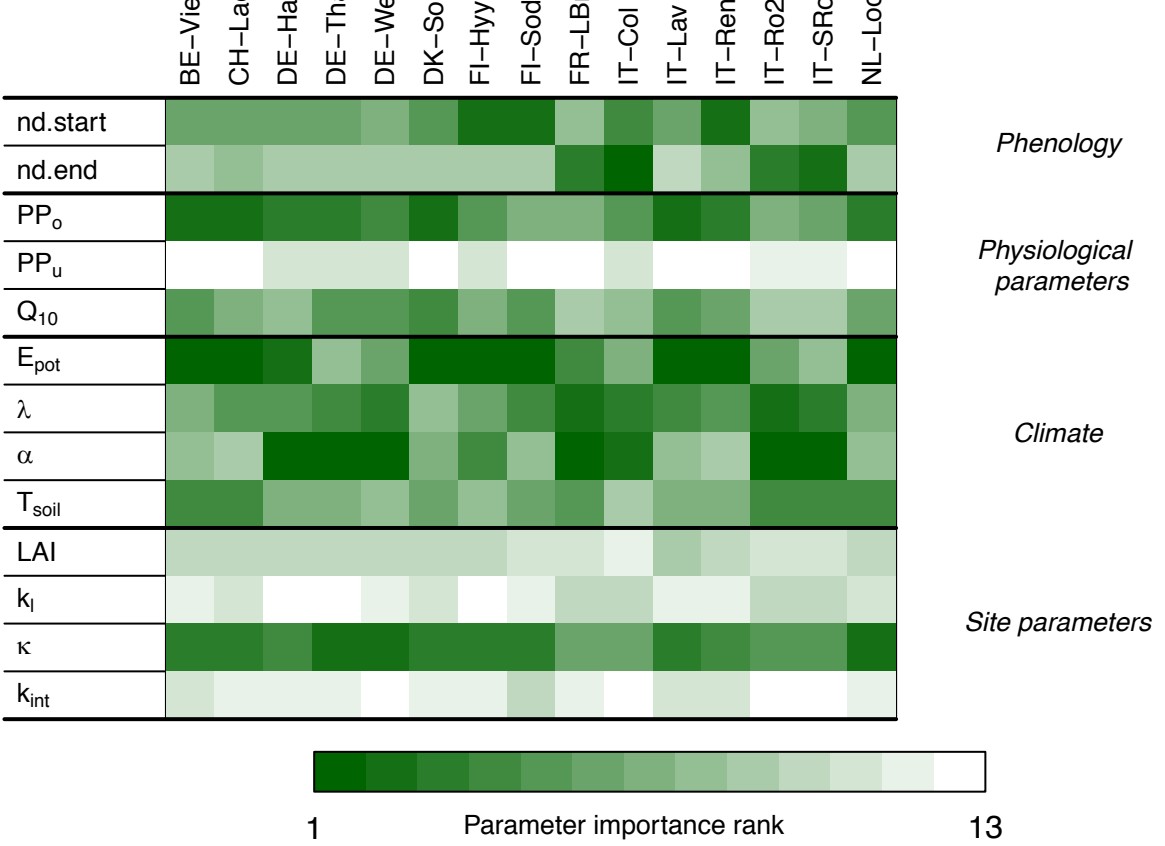

 **Figure 6: Importance rank of G-For10's input parameters. The darker the shade of green, the more influential a parameter is for the $S_r$ estimates at a given site. The model is most sensitive to perturbations of the physiological parameters of the overstory ($PP_o$) and potential evaporation ($E_{pot}$) at the cold and temperate sites, whereas variations in mean precipitation intensity ($\alpha$) become more important at Mediterranean sites. Also, variations in soil water holding capacity $\kappa$ consistently have a medium to high rank.**

The analysis described above does not indicate the sensitivity of $S_r$ estimates to the different parameters across their entire possible range, but only how perturbations of the parameter values given in Tables 1, 2 and 4 contribute to the uncertainty of $S_r$ estimates. In addition, it is also worthwhile to explore how site and vegetation parameters impact $S_r$ predictions under a given climate. Figure 7 shows how varying LAI (Fig. 7 a)), soil water holding capacity $\kappa$ (Fig. 7 b)) and the plant physiological parameter $PP_o$ (Fig. 7 c), see Eq. 15) affects $S_r$. Increasing LAI influences $S_r$ estimates by increasing the relative contribution of the overstory to total $S_r$, and by decreasing effective precipitation due to increasing interception evaporation (Eq. 10). The absolute effect of LAI varies greatly across the energy-limited sites. For example, shifting from a sparse canopy (LAI=1) to the current LAI of 9.6 at Lavarone increases $S_r$ by 100 mm, while $S_r$ is totally insensitive to LAI

at Loobos. The drier sites, represented by Roccarespampani on Fig. 7, show a low sensitivity of $S_r$ to LAI. Figure 7 b) shows the effect of varying $\kappa$ on $S_r$ and effective rooting depth of the overstory $Z_e$. At all sites, the optimal rooting depth decreases with increasing $\kappa$. However, for $S_r$, this is more than offset by the higher water holding capacity, so that $S_r$ increases with increasing $\kappa$. In Figure 7 c), $S_r$ was calculated with the vegetation parameter $PP_o$ ranging from half its standard value (0.0001) to double. As higher values of $PP_o$ represent a higher cost of roots, $S_r$ decreases with increasing $PP_o$ for a given soil and climate. Also here, the sensitivity of $S_r$ to $PP_o$ varies greatly across sites, with e.g. a halving of $PP_o$ leading to an increase in $S_r$ of 50 mm at Loobos and Roccarespampani, and over 100 mm at Lavarone. As shown in the inset of Fig. 7 c), the effect of $PP_o$ on $S_r$ (i.e. the difference between $S_r$ estimated with $PP_o = 0.00005$ and $PP_o = 0.0002$) increases with increasing $\kappa$.

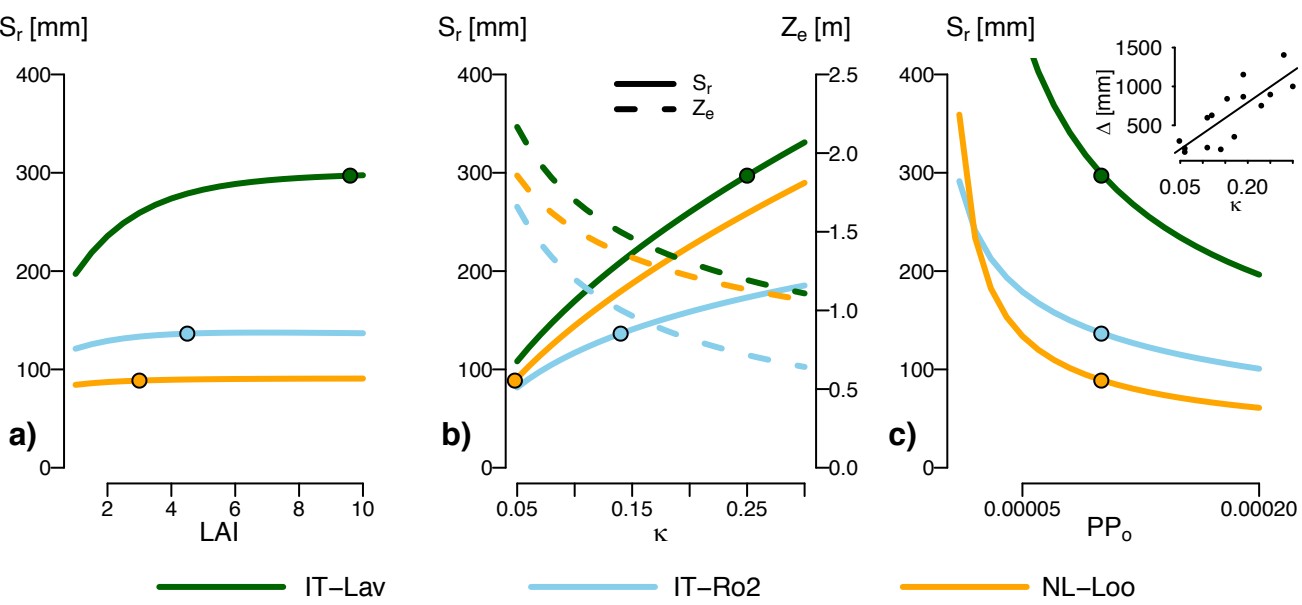

**Figure 7: Effect of three parameters on G-For10 predictions of $S_r$, for three contrasting sites. All other parameters are set to their standard or site-specific value. The dots show the $S_r$ estimates obtained with standard configuration (Table 2). (a) Change in $S_r$ as a function of LAI. As LAI increases, so does the contribution of the overstory to total $S_r$. Due to the differences in parameterization (Table 1) and water uptake model, the rooting depth for the overstory is always greater than for the understory. This effect is greatest at mesic sites with high soil water holding capacity, such as Lavarone. (b) Change in $S_r$ (solid lines) and overstory effective rooting depth $Z_e$ (dashed lines) as a function of soil water holding capacity $\kappa$. While an increase in $\kappa$ leads to shallower roots (decreasing $Z_e$), the effect on $S_r$ is inverse (increasing $S_r$). (c) Change in $S_r$ as a function of the parameter $PP_o$, which summarizes vegetation properties. The sensitivity of $S_r$ to $PP_o$ varies greatly among the sites, as e.g. halving the standard value of 0.0001 leads to an increase in $S_r$ of 50 mm at Loobos and Roccarespampani, and over 100 mm at Lavarone. The inset shows the relationship between $\kappa$ and the difference between $S_r$ calculated at $PP_o = 0.00005$ and $PP_o = 0.0002$ ($R^2=0.66$, p=0.0002).**

## 3.5 Effect of $S_r$ estimates on model performance

Figure 8 shows the $KGE_{evap}$ (Fig. 8 a)) and $KGE_{REW}$ (Fig. 8 b)) scores obtained at each site during the validation period for the three sets of validation runs (see Table 3). As described in Sect. 2.2.2, the only difference between these runs is the value of $S_r$. For $KGE_{evap}$, there is little difference between the three parameterizations, with the exception of San Rossore, Roccarespampani and Loobos (where model performance is worse with the modeled $S_r$), as well as Collelongo (where the model performs better when G-For08 or G-For10 estimates are used). For $KGE_{REW}$, the scores are generally lower, the spread higher, and the differences between the three sets of runs are more pronounced at some stations. At Roccarespampani and San Rossore, the calibrated parameter sets yield median $KGE_{RWE}$ scores of 0.4 and 0.65, respectively, while most of the runs including modeled $S_r$ obtained scores of zero or less. At Loobos, all three parameterizations performed badly, with the median of scores below zero in all three cases. The greatest difference between G-For08 and G-For10 occurs at Lavarone, where the median score for G-For10 is at 0.35, which is somewhat less than the median of the scores for the calibrated parameter sets (0.4), whereas the scores for all G-For08 runs are below zero. The other stations show some slight differences between the three sets of runs, but no consistent pattern is apparent.

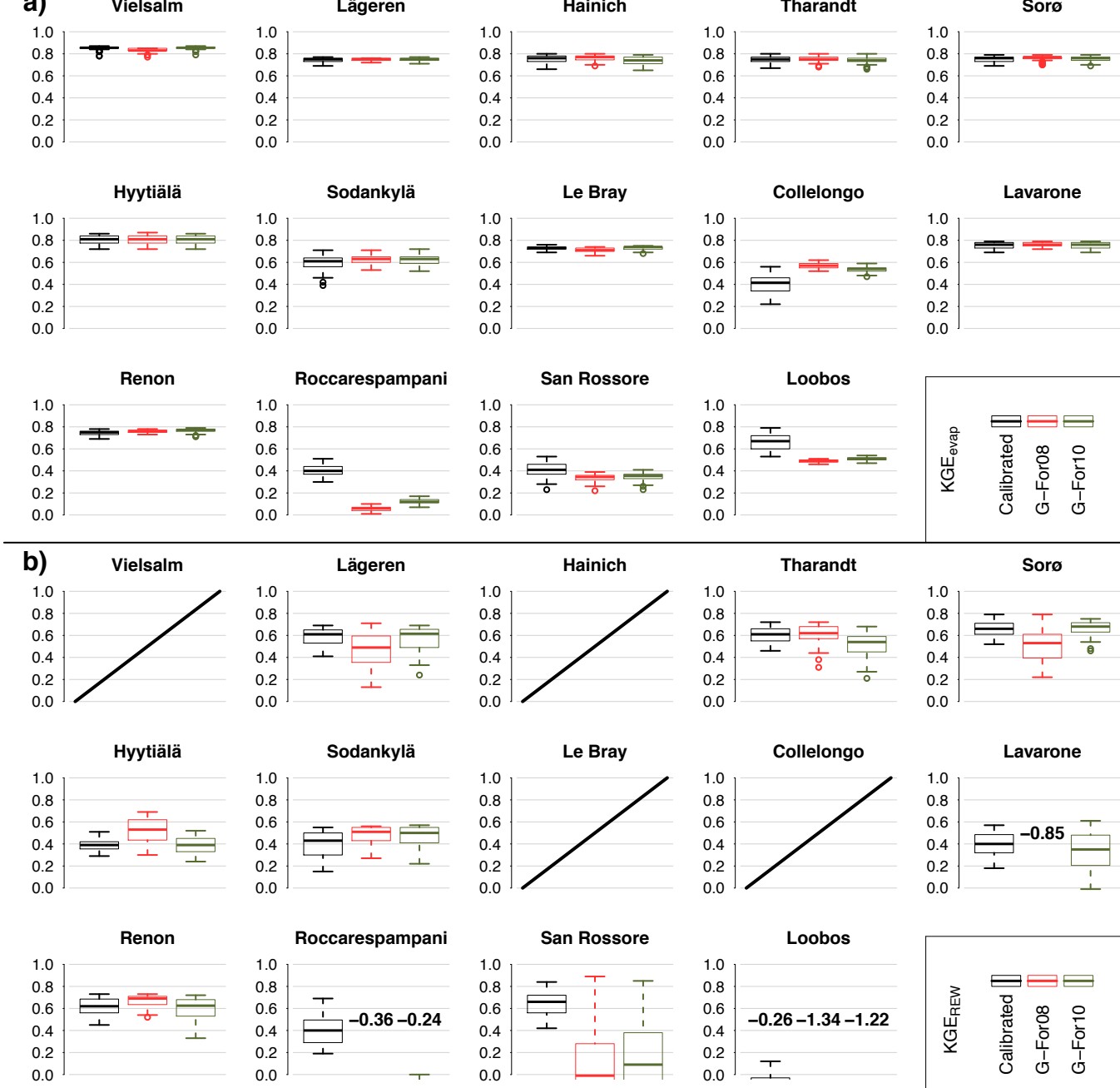

**Figure 8:** KGE scores obtained during the validation, for total evaporation $E_{tot}$ (a) and relative extractable soil moisture *REW* (b). The boxes in each plot correspond to the three sets of validation runs listed in Table 3 (parameter sets derived from calibration, and their two variants with $S_r$ replaced with the value from the G-For08 and G-For10 models). The center of the boxes represents the median of KGE scores, and the lower and upper bounds of the boxes show the first and third quartiles, respectively. The whiskers extend to the furthest values within 1.5 times the interquartile range (IQR) from the box bounds. Where the majority of KGE scores are below zero, the median is printed on the plot. No validation runs were done for Wetzstein, due to the short data record. Also, no soil moisture time series were available for validation at the four sites for which the graphs are crossed out on (b).

## 4 Discussion

### 4.1 Calibrated $S_r$ and their uncertainty

The goodness-of-fit scores obtained by FORHYTM during validation (Fig. 8) give an indication of the reliability of the calibrated $S_r$ estimates and, more generally, on the suitability of this model structure to simulate local water balance under various conditions. For comparison, Chaney et al. (2016), who also calibrated an evaporation routine against half-hourly eddy covariance data using $KGE$, obtained a median score of 0.73 after parameter optimization. Sprenger et al. (2015) obtained $KGE$ scores ranging from 0.43 to 0.8 for soil moisture time series. The scores obtained here for $E_{tot}$ range from 0.66 to 0.87 for temperate and boreal sites, and from 0.46 to 0.58 at Mediterranean sites. For the REW time series, the sites with scores below the range cited above are Roccarespampani and Loobos. These results suggest that FORHYTM is able to reproduce the local water balance at most temperate and boreal sites, but that its predictive ability is limited under Mediterranean conditions. By extension, this gives confidence that the calibrated $S_r$ are representative for the actual site conditions, at least at the temperate and boreal sites. An exception within the temperate sites is Loobos, where the performance of FORHYTM for REW was worst (0.12). Possible reasons for poor model performance at certain sites are discussed in Appendix B.

A sensitivity analysis conducted in a previous study (Speich et al., 2018) highlighted the high importance of $S_r$ for long-term water balance modeling, and the additional analysis conducted here shows that this also holds for modeling at finer timescales, and that $S_r$ is an important factor for model performance (Fig. 3, Table B2). Together with the validation results, this suggests that the calibration of FORHYTM is an acceptable method to estimate $S_r$ at most sites. However, the calibrated $S_r$ values are still subject to considerable uncertainty. This is partly due to the influence of other parameters, which leads to equifinality (Beven, 1993; Chaney et al., 2016), i.e. the existence of several parameter sets that yield equally good results. One way to account for this is to represent parameter values as a distribution or, as done here, a range, instead of a unique value. Another source of prediction uncertainty is the uncertainty in the input and calibration data. For example, the micrometeorological measurements at the FLUXNET sites may contain gaps (due e.g. to instrument failure), in which case the value has been estimated with gap-filling methods or downscaled from external datasets, as part of the FLUXNET data processing workflow. Latent heat flux measured with the eddy covariance technique is subject to various types of uncertainty from different sources (Richardson et al., 2012). Some of these errors are of random nature, with an expected value of zero. Such random errors might decrease the level of agreement between modeled and measured fluxes, but are unlikely to introduce any biases into the calibrated $S_r$ estimates. Other measurement errors are of systematic nature and may cause a consistent under- or over-estimation of fluxes, which impacts the calibrated parameter values. While various techniques are applied by the data providers to reduce these uncertainties (Baldocchi et al., 2001; Mauder et al., 2013), they cannot be fully eliminated. For measured soil moisture, on the other hand, the issues are mostly linked to horizontal and vertical heterogeneity of soil properties and of water content and movement (Allaire et al., 2009; Coenders-Gerrits et al., 2013).

## 4.2 Behavior of the optimal rooting depth models

### 4.2.1 Differences between G-For08 and G-For10

As seen in Table 4 and Fig. 5 b), the difference between G-For08 and G-For10 estimates varies greatly among sites. Greater differences are found at energy-limited sites, and the difference also increases with increasing water holding capacity $\kappa$.
Guswa (2010) also found that the G10 model always leads to deeper roots, and that the difference between both model versions was most pronounced under energy-limited conditions. This is explained by the differing goals of vegetation types using an intensive (e.g. grass) and a conservative water uptake strategy (e.g. trees). Plants with an intensive water-use strategy maximize the capturing of incoming precipitation by quickly depleting the soil moisture reservoir, so that a higher fraction of the next precipitation event is available to the roots. With a conservative strategy, the soil dries out less quickly, so that for a given rooting depth, a higher fraction of the next precipitation event would run off. Therefore, deeper roots allow the vegetation to retain a higher fraction of precipitation. In addition, under energy-limited conditions, a large reservoir ensures that soil moisture remains high. If the transpiration rate depends on soil moisture, as in the G10 model, this allows the vegetation to maximize transpiration and carbon intake. On the other hand, if transpiration always occurs at the potential rate, as in G08, there is less benefit in maximizing soil moisture. Therefore, with the G08 model, optimal rooting depth decreases as $W$ increases above one, whereas G10 is relatively insensitive to changes in $W$ above one, especially when the mean precipitation intensity is high (Guswa, 2010). In three of the sites used in this study, precipitation is substantially larger than potential evaporation: Lavarone, Renon and Lägeren (Fig. 4 a); Table 5). These sites are also characterized by high mean precipitation intensity (Fig. 4 b)). As seen in Sect. 3.3, the difference between G08 and G-For also depends on water holding capacity $\kappa$, as sites with a high $\kappa$ show a greater difference than sites with a similar climate but a lower $\kappa$. Therefore, decreasing $\kappa$ may have a similar effect as a shift to drier conditions: an effective rooting depth that maximizes transpiration may not be worthwhile if the soil can store little water. Under greater water availability (due to climatic or edaphic factors), effective rooting depth is more sensitive to changes in LAI, plant properties and uptake strategy.

### 4.2.2 Parameter sensitivity and uncertainty of the G-For10 model

The distributions of the $S_r$ estimates obtained during the uncertainty analysis at each site (Sect. 3.4; Fig. S-16) indicate that the results of G-For10 can vary substantially if the input parameters are varied within a relatively narrow range. Figure 6 shows the ranking of parameters with regard to their contribution to the uncertainty of G-For10 estimates at each site. As seen in Sect. 3.4, $S_r$ estimates at temperate and cold sites are generally most sensitive to perturbations of potential evaporation and of the summary plant parameter $PP_o$, whereas perturbations of precipitation, especially the average precipitation depth $\alpha$, are more important at Mediterranean sites. The greater sensitivity to precipitation under water-limited conditions is consistent with the observations of Schenk and Jackson (2002). Canopy characteristics like LAI and $k_l$ are of little importance, whereas soil water holding capacity $\kappa$ is at least of intermediate importance at all sites. Indeed, as can be seen on Fig. 7 b), the effect of $\kappa$ on $S_r$ is quasi-linear, so that a 20% change in $\kappa$ has a similar effect, regardless of the

standard value. Variations in the start of the growing season are important at boreal and high-elevation sites, whereas variations in the end of the growing season are more important at Mediterranean and maritime sites. The spring phenology model of Kramer (1996), which was used in this study, was parameterized on trees in Germany and the Netherlands, and might not be accurate under different climatic conditions. Likewise, the criterion to determine the end of the growing season (Sect. 2.1.3) is entirely arbitrary. It should thus be possible to better constrain the G-For models by using site-specific phenology models.

Figure 7 a) shows the dependence of $S_r$ on LAI for energy-limited and water-limited sites. Increasing LAI causes the contribution of overstory $S_r$ to total $S_r$ to increase (Eq. 8). As discussed in Sect. 4.2.1, rooting depth estimates of the G10 model are consistently lower than or equal to G08 estimates. Furthermore, differences in vegetation parameters between overstory and understory (Table 1) also lead to a greater rooting depth for the overstory. Under energy-limited conditions (exemplified by Lavarone), $S_r$ increases with increasing LAI, up to the point where the curve flattens off, and further changes in LAI have little effect. From an ecological perspective, this is in line with expectations: a closed forest has a greater demand for transpiration than a sparse forest, so that a larger reservoir is necessary. Changes in LAI within an already dense forest, however, have little additional effect on potential transpiration (Granier et al., 1999). Also, studies comparing forest stands in different developmental stages showed that rooting properties varied little once canopy closure was reached (Kalliokoski et al., 2010). Under drier conditions, $S_r$ is much less sensitive to changes in LAI. This is also the case at temperate sites with low water holding capacity, such as Loobos. This is in line with the discussion in Sect. 4.2.1: the effect of water uptake strategy is greatest where ecosystems are less limited by water availability, due to climatic or edaphic conditions.

The increase of effective rooting depth with lower $\kappa$ (Fig. 7 b)) is consistent with the model results of Collins and Bras (2007). They note, however, that deeper roots are also to be expected in very fine soils (high $\kappa$), if macropores and groundwater are present. Neither G08/G10 nor the model of Collins and Bras (2007) accounts for these factors. The comparison between the $Z_e$ and $S_r$ curves shows that, although the optimal rooting depth becomes shallower with higher $\kappa$, the storage volume increases. Therefore, all else being equal, plants need to spend less carbon on roots on soils with a higher $\kappa$, and can transpire more (and assimilate more carbon) with roots at their optimal level.

The vegetation parameter $PP_o$ is an influential parameter for $S_r$ estimates, particularly under mesic conditions (Fig. 6) and on soils with a high $\kappa$ (Fig. 7 c)). While the generic parameterization used here is based on values reported in the literature (Table 1), the plant parameters that make up $PP_o$ may vary across species. For example, the root morphological parameters differ between broadleaves and conifers, with a markedly higher specific root length $L_r$, and a tendency towards higher root length density $D_r$, in the former (Kalliokoski et al., 2010; Withington et al., 2006). If the relative difference in $L_r$ is higher than that in $D_r$, this would mean that broadleaves have a tendency to form deeper roots than conifers. The variables related to the plant's carbon budget can also be expected to vary across species groups. Typically, species with a high degree of shade tolerance tend to have a higher water-use efficiency and lower respiration rates, and vice-versa (Polster, 1950; Valladares

and Niinemets, 2008). Adjusting $PP_o$ accordingly would lead to larger $S_r$ estimates for more shade-tolerant species. Also, root respiration rates, as well as the temperature coefficient $Q_{10}$ (not reflected in $PP_o$) may vary with climatic conditions (Burton et al., 2002). In addition, water-use efficiency and respiration rates vary seasonally (Larcher, 2001). All these factors make it difficult to constrain $PP_o$, which may be seen as a large source of uncertainty for this model.

### 4.2.3 Differences between calibrated $S_r$ and G-For estimates

Among the sites with the greatest difference between modeled and calibrated $S_r$ are the Mediterranean and maritime sites Roccarespampani, San Rossore, Collelongo and Le Bray (ellipse 1 on Fig. 5 a)). As noted before, the performance of the water balance model was relatively low at these sites, which also reduces confidence in the calibrated $S_r$. Another possible explanation for the mismatch at Roccarespampani is that this site is a coppice, and thus its trees are very young (11 years at the beginning of measurements (Papale et al., 2015)). Therefore, the forest may be far from a steady state, making optimality-based model predictions less reliable. Also, coppiced systems tend to have a high root:total biomass fraction (Deckmyn et al., 2004), which might further explain the mismatch between modeled and calibrated $S_r$. At San Rossore, the presence of a water table at 1 to 2 m below ground (Papale et al., 2015) is another factor that may influence the rooting strategy of the vegetation. Indeed, the case where a plant sends deep roots in search of a water table is not covered by G08/G10 (Guswa, 2008). Glenz (2005) proposed a modeling strategy for those cases. Another explanation for the differences between calibrated and modeled $S_r$ at these sites may be found in the way precipitation is represented (Eq. 10). The use of precipitation frequency and main intensity during the growing season presupposes that the local water balance depends on complete or partial rewetting of the soil by rainfall events occurring during the growing season. This assumption is reasonable under cool temperate conditions. In Mediterranean climates, precipitation events are often distributed very unevenly over a year, with only a small amount of rainfall during the summer half year. The seasonality of precipitation is thus more important for the vegetation than the distribution of rainfall events during the summer half-year. For these cases, Guswa (2008) suggests setting a very low frequency (e.g. $\lambda = 1/180$), and a very high mean precipitation intensity, to reflect the fact that the water available to plants in summer mostly falls in winter and is stored in the soil. The effective amount of water available at the beginning of summer depends on soil hydrology during the wet season, and might be estimated using a model like the one of Porporato et al. (2004). As a first approximation, the plausibility of this alternative approach was tested by setting $\lambda$ and $\alpha$ to the values proposed by Guswa (2008) ($\lambda = 1/180$ events per day, $\alpha = 500$ mm) at the four Mediterranean and maritime sites, while keeping all other factors unchanged. This resulted in much higher estimates of $S_r$ than in Table 4, with differences ranging between 62 mm (San Rossore) and 545 mm (Collelongo). While these new $S_r$ estimates are still quite far from the calibrated values, this shows that the precipitation model has a great influence on $S_r$ estimates and may need to be adapted before the G08 and G10 models can be applied under such climates.

Three of the sites where modeled and calibrated $S_r$ differ most (San Rossore, Loobos and Le Bray – ellipse 2 on Fig. 5 a)) are pine stands growing on sandy soils. *Pinus* roots often show a high degree of adaptation to soil conditions (Hacke et al.,

2000; Kutschera and Lichtenegger, 2002). It is then conceivable that the carbon cost of roots decreases in coarser soils, allowing the trees to develop deeper roots than on finer soils. However, halving the vegetation parameter $PP_o$ (i.e. decreasing the cost of roots) only leads to a modest increase in $S_r$ at Loobos (Fig. 7c)). As discussed in Sect. 4.2.2, the plant parameters summarized in $PP_o$ are poorly constrained, and it is difficult to determine a realistic range for $PP_o$. It is thus not possible to

conclude how well G08 and G10 capture the rooting behavior of pines on sandy soils. It is also possible that the reason for deeper rooting systems in sandy soils is the avoidance of cavitation (Hacke et al., 2000), which is a different objective than the carbon budget optimization assumed here. Furthermore, a possible strategy of pines on coarse soils is to develop a highly heterogeneous rooting system, comprising both deep taproots and preferential root development in patches with higher humidity and nutrient supply (Kutschera and Lichtenegger, 2002). In such cases, the simplified representation of $S_r$ as the

product of rooting depth and $\kappa$ might not be valid.

G08 and G10 estimate rooting depth based on water use optimization only, explicitly neglecting other constraints. According to Kutschera and Lichtenegger (2002), two of the main limitations to rooting depth are oxygen deficiency and low soil temperature. The latter applies primarily in temperate and cold climates, and may be amplified by high soil moisture content. In the temperature-dependent formulation proposed by Yang et al. (2016) and adopted here, low temperatures even promote

root growth by decreasing the respiration costs. Norway spruce (*Picea abies*) is particularly sensitive to these factors, often causing it to form shallow rooting systems (Kutschera and Lichtenegger, 2002). This offers an explanation why the optimality-based $S_r$'s are higher than the calibrated values at two of the three spruce sites. Indeed, the difference is much larger at the high-elevation site Renon (1794 m asl) than at Tharandt (390 m asl) and Wetzstein (703 m asl) (ellipse 3 on Fig. 5 a)), which supports the hypothesis that the discrepancy is linked to temperature.

**4.3 Theoretical considerations**

The G08 and G10 models are based on the assumption that plants dimension their rooting system to optimize their carbon budget. This involves processes taking place at the scale of an individual plant. However, these models were applied here at the scale of a community, thus neglecting any form of interactions between individuals. Various types of belowground interactions between forest trees have been reported, ranging from competition to facilitation (González de Andrés et al.,

2017), and these interactions may alter root morphology and distribution (Bolte and Villanueva, 2006). Likewise, the interactions between overstory and understory roots are represented here in a simplistic way, neglecting any form of competition. A somewhat related scaling issue arises from the fact that the model neglects the spatial heterogeneity of above- and belowground vegetation and soil properties. Both may influence the spatial distribution of soil moisture (Coenders-Gerrits et al., 2013), and it is unclear to what extent their variability influences the average rooting depth over a forest stand.

Such scaling issues are common in environmental modeling (Blöschl and Sivapalan, 1995), and the good agreement between calibrated $S_r$ and G-For10 results suggests that the model may be applied at the stand scale despite the simplifications discussed in this paragraph.

The only difference between G08 and G10 is the function relating mean transpiration to rooting depth. While G08 assumes no transpiration regulation until soil moisture is fully depleted, G10 assumes that transpiration is linearly reduced as soon as soil moisture is no longer at saturation. As noted by Guswa (2010), these are two extreme assumptions, whereas most vegetation types show an intermediate behavior. Indeed, the reduction of transpiration when soil moisture is below a certain

threshold is well documented for forests (Granier et al., 1999) and implemented in many dynamic models (Bergström, 1992; Granier et al., 1999; Zappa and Gurtz, 2003). Any equation relating transpiration to rooting depth could be used in Guswa's model. An equation reflecting an intermediate strategy would probably lead to results between G08 and G10 estimates, with the greatest effects where $S_r$ was shown to be most sensitive to water uptake strategy, i.e. at mesic sites with high water holding capacity.

By making rooting depth dependent on climatic variables, the G08 and G10 models may serve as a tool to analyze how future climate change may affect water storage. In Europe, climate models predict an increase in mean annual temperature in all regions (e.g. Jacob et al., 2014), which has the effect to increase evaporative demand. An increase in annual total precipitation is expected in cold and temperate regions, and a decrease is expected in Mediterranean regions (Giorgi and Lionello, 2008; Jacob et al., 2014). Furthermore, temporal rainfall patterns are expected to change, with a tendency towards

more intense events and longer dry spells (Jacob et al., 2014). As the relative magnitude of change is greater for temperature than for precipitation, these changes are likely to cause drier conditions (lower $W$) over the range of sites considered here. In regions that are currently energy-limited, this would cause the G08/G10 models to predict an increase in rooting depth, especially for the intensive water-use strategy (G08). This increase would be further enhanced by the lower frequency of rainfall events. All else equal, this would mean that a greater fraction of precipitation is transpired, thus reducing streamflow.

In Mediterranean regions, the models would predict a decrease in rooting depth, as the wetness index would be even further from 1. It is also likely that future drought events impact stand productivity and tree vitality (Granier et al., 2007), eventually causing a decrease in aboveground stand density. LAI, which represents aboveground vegetation properties in this implementation of the models, was shown to be a rather insensitive parameter. An exception is the lower range of LAI under energy-limited conditions (Fig. 7 a)). This means that a shift from a closed to a sparse forest would cause a substantial

decrease in modeled storage capacity whereas moderate changes in stand density would have little effect on storage capacity estimates. Storage capacity may also be altered as a result of changing species composition, as species might differ in their physiological properties (Sect. 4.2.2) and preferential rooting patterns (Sect. 4.2.3). Furthermore, climate change is likely to alter snow storage and growing season length, which both may impact the estimates of the G08/10 models (Yang et al., 2016).

**4.4 Implications for model development**

**4.4.1 Effect of different $S_r$ estimates on water-balance model performance**

The motivation for testing the G-For models is to assess whether they may be implemented in dynamic (eco)hydrological models. Figure 8 shows the $KGE$ scores obtained with the dynamic water balance FORHYTM during the validation period at

each site, with three different $S_r$ estimates: calibrated values, and G-For08 and G-For10 estimates. For evaporation, all versions give similar $KGE$ scores, except at Roccarespampani, San Rossore, Loobos and Collelongo. At all of these sites, $KGE_{evap}$ is lower than at the others for all three parameterizations. At the three former sites, G-For08 and G-For10 parameterizations lead to worse performance, while at Collelongo, the versions with modeled $S_r$ perform better than with the calibrated values. At these four sites, modeled $S_r$ was smallest relative to the calibrated values (Table 4; Fig. 5 a)). On the

other hand, where G-For10 estimates are larger than calibrated $S_r$ (Renon, Tharandt, Hainich), all parameterizations perform equally well. For $KGE_{REW}$, the differences between the three parameterizations are larger, suggesting that simulations of soil moisture dynamics are more sensitive to $S_r$. Also here, all parameterizations perform equally well at Renon, despite great differences in $S_r$. Together with the relationship between $S_r$ and $KGE_{AVG}$ (Fig. 3), this suggests that an underestimation of $S_r$ has greater effects than an overestimation.

The most striking difference between the parameterizations with G-For08 and G-For10 occurs at Lavarone, where the G-For10 parameterization performs almost equally well as the calibrated parameter sets, whereas none of the runs using the G-For08 parameterization obtained a $KGE_{REW}$ score above zero. As Lavarone is the site with the greatest difference between G-For08 and G-For10 estimates of $S_r$ (Fig. 5 b)), this would suggest that G-For10 is more suitable than G-For08 to estimate the rooting storage capacity of energy-limited forests. However, at other sites with a great difference between G-For08 and

G-For10 estimates of $S_r$ (e.g. Hyytiälä, Sorø, Lägeren), all parameterizations perform similarly well. It is thus not possible to conclude whether G-For10 is a better model than G-For08 for the range of sites considered in this study.

**4.4.2 Alternative methods for $S_r$ modeling**

As mentioned in the introduction, an alternative approach to parameterize $S_r$, based on the return period of soil moisture deficits (hereafter referred to as the mass balance approach; de Boer-Euser et al., 2016; Gao et al., 2014; Wang-Erlandsson et

al., 2016), was recently used to generate time-varying estimates of $S_r$ for a dynamic hydrological model (Nijzink et al., 2016). Due to the relative novelty of both approaches in dynamic modeling, it is worthwhile to compare their properties. The mass balance approach assumes that the vegetation dimensions its rooting system so that it can withstand soil droughts with a certain return period (e.g. 20 years; Nijzink et al., 2016). It requires time series of daily precipitation and transpiration. The cumulative sum of transpiration minus precipitation is calculated daily, and the greatest value for each year is recorded.

Storage capacity $S_r$ is estimated from these maximal annual deficits using extreme value statistics.

Compared to the method presented in this paper, the requirements for data, and especially parameter values, are much lower for the mass balance approach. Considering the high uncertainty of the G-For model parameters and its propagation to model

results, as discussed in Sect. 4.2.2, this is an advantage of the mass balance approach. On the other hand, in its current form, the mass balance approach must be calculated a priori. This hinders its application to cases for which measurements are not available, e.g. under future climate change scenarios. By contrast, as many of the inputs for G-For are typically used or simulated in a hydrological model, it can be directly integrated into the model formulation, and updated e.g. based on rolling long-term averages of the climate statistics. Interestingly, both approaches consider different aspects of temporal variability. The G08/G10 models use long-term averages, and account for rainfall intermittency by describing rainfall with the parameters $\lambda$ (frequency) and $\alpha$ (mean intensity). In the mass balance approach, rainfall intermittency is reflected in the annual deficit calculations (less frequent precipitation events will cause greater deficits). The seasonality characterizing Mediterranean conditions, discussed in Sect. 4.2.3, would be properly captured with such an approach. Additionally, the mass balance approach takes into account the inter-annual variability of climatic variables by considering extreme value statistics. The models examined here, on the other hand, do not account for this. This is a drawback of these models, as climatic extremes may have a greater impact on physiological processes than changes in mean values (Reyer et al., 2013).

### 4.4.3 Potential applications of G-For in hydrological modeling

As discussed in Sect. 4.4.1, using G-For estimates of $S_r$ in the water balance model FORHYTM led to similar performance as with calibrated values in most temperate and cold locations. This suggests that G-For could be implemented in a hydrological model under such conditions. At these locations, the $S_r$ estimates respond to changes in climate and above-ground vegetation structure in a way that is in line with ecological theory (4.2.2). The example of Collelongo (Appendix B) shows that non-stationarity of climatic conditions can affect the transferability of calibrated parameter values. Thus, implementing a time-varying formulation of $S_r$ using G-For could greatly increase the credibility of climate impact projections (Montanari et al., 2013; Savenije and Hrachowitz, 2017). Furthermore, the numerical approximation presented here for G10 (Sect. 2.1.2) facilitates the implementation of the model. At some sites dominated by Norway spruce, modeled $S_r$ is much greater than the calibrated value. This might be explained with ecological processes not accounted for by the G-For model (Sect. 4.2.3). Therefore, the model could be improved by specifying a penalty or limitation in cases of low soil temperature or oxygen stress. Besides full integration in a dynamic model, G-For can also be used to constrain model calibration, thus contributing to reduce parameter uncertainty.

On the other hand, using G-For estimates at Mediterranean sites, as well as at Pine-dominated sites on sandy soils, led to lower performance of the dynamic model. As discussed in Sect. 4.2.3 and Appendix B, this might by caused by various factors. Accounting for species- and site-specific variations in vegetation parameters, which control the carbon cost of roots, might improve the model, although it cannot be determined how realistic such variations are. Also, the concept of $S_r$ used in this paper might be inappropriate for very coarse soils (Sect. 4.2.3). Finally, the low performance at these sites may indicate that the dynamic water balance model used here is inappropriate for such conditions. The results of this study do not permit a definite conclusion on the reason for the poor results of G-For at these sites. Therefore, further research is needed before G-For or similar models can be applied to such conditions.

Many of the inputs required by G-For can be easily calculated from the inputs of a typical hydrological model. For example, the climate statistics in this study were calculated from the meteorological variables typically required for the Penman equation (Penman, 1948). Climatic factors contributed significantly to the uncertainty of $S_r$ estimates (Fig. 6). As trees adapt their rooting systems to long-term climate (Wang-Erlandsson et al., 2016), it is advisable to use a large time window to calculate the climate statistics for G-For. This also reduces the uncertainty associated with the estimates of climate statistics. Another important source of uncertainty is soil water holding capacity $\kappa$. Due to its high horizontal and vertical heterogeneity, this parameter is often poorly constrained. However, the development of spatially coherent datasets at relatively fine resolution is an area of active research (e.g. Tóth et al., 2017). The plant parameters of the G-For model also contribute greatly to uncertainty (Fig. 6), and can be difficult to constrain. Due to the reasonable G-For10 estimates obtained at a majority of temperate and cold sites, we recommend to use the generic parameterization used in this paper in the absence of better information.

## 5 Conclusion

In this study, we assessed the potential of an optimality-based rooting depth model to parameterize rooting zone storage capacity $S_r$ in temperate forests. This model is based on the assumption that plants dimension their rooting systems in a way that maximizes their carbon budget. We compared two versions of the model, differing in their assumptions regarding plant water uptake strategy. As observations of rooting profiles are scarce and performed at a spatial scale much smaller than the typical discretization unit in models, it was not possible to compare the results of the rooting depth functions with direct measurements. Instead, $S_r$ estimates were obtained by calibrating a water balance model against observations of latent heat flux and soil moisture dynamics at 15 eddy covariance stations. Then, the impact of using modeled $S_r$ estimates on the performance of the dynamic model was assessed during a validation period.

The results showed that the level of agreement between calibrated and modeled $S_r$ varied widely across climates and forest types. In a majority of cold and temperate sites, calibrated and modeled $S_r$ agreed relatively well. Accordingly, the dynamic water balance model performed equally well with calibrated and optimality-based $S_r$ at these sites. At sites dominated by Norway spruce, optimality-based $S_r$ was much higher than the calibrated value. However, there was little difference in the performance of the dynamic model, i.e. the model performed equally well with calibrated and with modeled $S_r$. This suggests that an overestimation of effective rooting depth has less effect on local water balance predictions than an underestimation at these sites. Nevertheless, $S_r$ estimates could be improved by including the effects of low soil temperatures and oxygen deficiency, which are not accounted for by the optimality-based models.

On the other hand, optimality-based $S_r$ were consistently much lower than calibrated values at the Mediterranean sites considered in this study. The same was the case for Pine-dominated sites on sandy soils. Accordingly, the water balance model performed substantially worse at these sites when optimality-based $S_r$ values were used. A possible explanation for these mismatches is that trees under these conditions follow strategies that differ from the carbon optimization objective

assumed by the model. For example, trees might rather minimize the risk of cavitation, create a buffer for extreme droughts or develop heterogeneous rooting systems depending on patterns of water and nutrients availability. An alternative explanation is that the representation of precipitation used in this study does not reflect the seasonality of precipitation, which is important in Mediterranean climates. However, due to the small number of sites concerned, it is not possible to determine with certainty the cause of these mismatches.

The results of this study indicate that this optimality-based parameterization of effective rooting depth has the potential to be used in dynamic (eco)hydrological model under cold and temperate conditions, either as a model component or as a way to constrain model calibration. On the other hand, the results obtained here do not warrant its application in Mediterranean climates and on very coarse soils. Further research is needed to determine the rooting strategies of trees under these conditions, and whether these strategies can be reconciled with the concept of a bulk rooting zone storage capacity.

## Appendix A: List of symbols

**Table A1: List of all symbols used in this paper. The variables marked with an asterisk are the ones used in the sensitivity and uncertainty analysis.**

| Symbol | Meaning | Units |
|---|---|---|
| *Rooting depth and rooting zone storage capacity* | | |
| $S_r$ | Rooting zone storage capacity | mm water depth |
| $Z_e$ | Effective rooting depth | mm |
| $Z_n$ | Number of average precipitation events that can be stored in the rooting zone | mm m$^{-1}$ |
| *Plant physiological parameters of the G08 and G10 models* | | |
| $w_{ph}$ | Photosynthetic water use efficiency (WUE) | mmol CO$_2$ cm$^{-3}$ water |
| $\gamma_{r,20}$ | Root respiration rate at 20 °C | mmol CO$_2$ g$^{-1}$ roots day$^{-1}$ |
| $Q_{10}$ | Temperature coefficient for root respiration * | - |
| $L_r$ | Specific root length | cm roots g$^{-1}$ roots |
| $D_r$ | Root length density | cm roots cm$^{-3}$ soil |
| $PP_o$ | Vegetation parameter, summarizing $w_{ph}, \gamma_{r,20}, L_r$ and $D_r$ for the overstory (Eq. 15) * | day$^{-1}$ |
| $PP_u$ | Vegetation parameter, summarizing $w_{ph}, \gamma_{r,20}, L_r, D_r$ and $f_{seas}$ for the understory (Eq. 16) * | day$^{-1}$ |
| *Climatic parameters of the G08 and G10 models* | | |
| $E_{pot}$ | Potential evaporation * | mm day$^{-1}$ |

| | | |
|---|---|---|
| $T_{pot}$ | Potential transpiration | mm day$^{-1}$ |
| $T_{pot,o}$ | Potential transpiration of the overstory | mm day$^{-1}$ |
| $T_{pot,u}$ | Potential transpiration of the understory | mm day$^{-1}$ |
| $\alpha$ | Mean rainfall intensity * | mm event$^{-1}$ |
| $\lambda$ | Frequency of rainfall events | events day$^{-1}$ |
| $P$ | Incoming precipitation * | mm day$^{-1}$ |
| $P_{eff}$ | Effective precipitation | mm day$^{-1}$ |
| W | Wetness index (= $P_{eff}/T_{pot}$ ) | - |
| $T_{soil}$ | Mean soil temperature during the growing season * | °C |
| $f_{seas}$ | Length of growing season | Fraction of a year |
| ndays.start, ndays.end | Perturbation of start and end dates of the growing season in the sensitivity analysis of G10 * | days |
| *Site-specific parameters of the G08 and G10 models* | | |
| LAI | Leaf Area Index * | m$^2$ m$^{-2}$ |
| $\kappa$ | Soil water holding capacity * | mm water depth mm$^{-1}$ soil depth |
| $k$ | Canopy light extinction coefficient * | - |
| $S_{int}$ | Canopy interception storage | mm |
| $k_{int}$ | Link between interception storage and LAI * | mm |
| *Calibration parameters of the dynamic water balance model FORHYTM* | | |
| $\beta$ | Shape coefficient of the soil moisture recharge function | - |
| $r_{s,min}$ | Minimum stomatal resistance | s m$^{-1}$ |
| $k_{soil}$ | e-folding time of the soil evaporation reduction function | days |
| $j_{vpd}$ | Exponent of the VPD-induced reduction of stomatal conductance | - |
| $l_{vpd}$ | Threshold for stomatal response to VPD | hPa |
| $k_{int}$ | Link between interception storage and LAI | mm |

## Appendix B: FORHYTM model description and validation results

The dynamic water balance model FORHYTM model consists essentially of a coupling between the dual-source transipration and soil evaporation routine of Guan and Wilson (2009) and a soil water balance routine widely used in semi-conceptual hydrological models (Bergström, 1992; Zappa and Gurtz, 2003). Figure B1 a) gives an overview of the water

fluxes simulated in FORHYTM. The scheme of Guan and Wilson (2009) assumes an interaction between the energy fluxes between overstory and understory, while accounting for the difference in evaporation between inter-canopy and sub-canopy understory parts. In this routine, available energy, represented by net radiation, is partitioned between overstory and understory/soil using Beer's law (see Eq. 8). Potential transpiration and soil evaporation are then calculated using Penman-

5 Monteith-type equations and scaled according to fractional canopy cover. Incoming precipitation first fills an interception reservoir, whose size ($S_{int}$) is related to LAI through an empirical relationship proposed by Menzel (1997) and Vegas Galdos et al. (2012) (see Eq. 11).

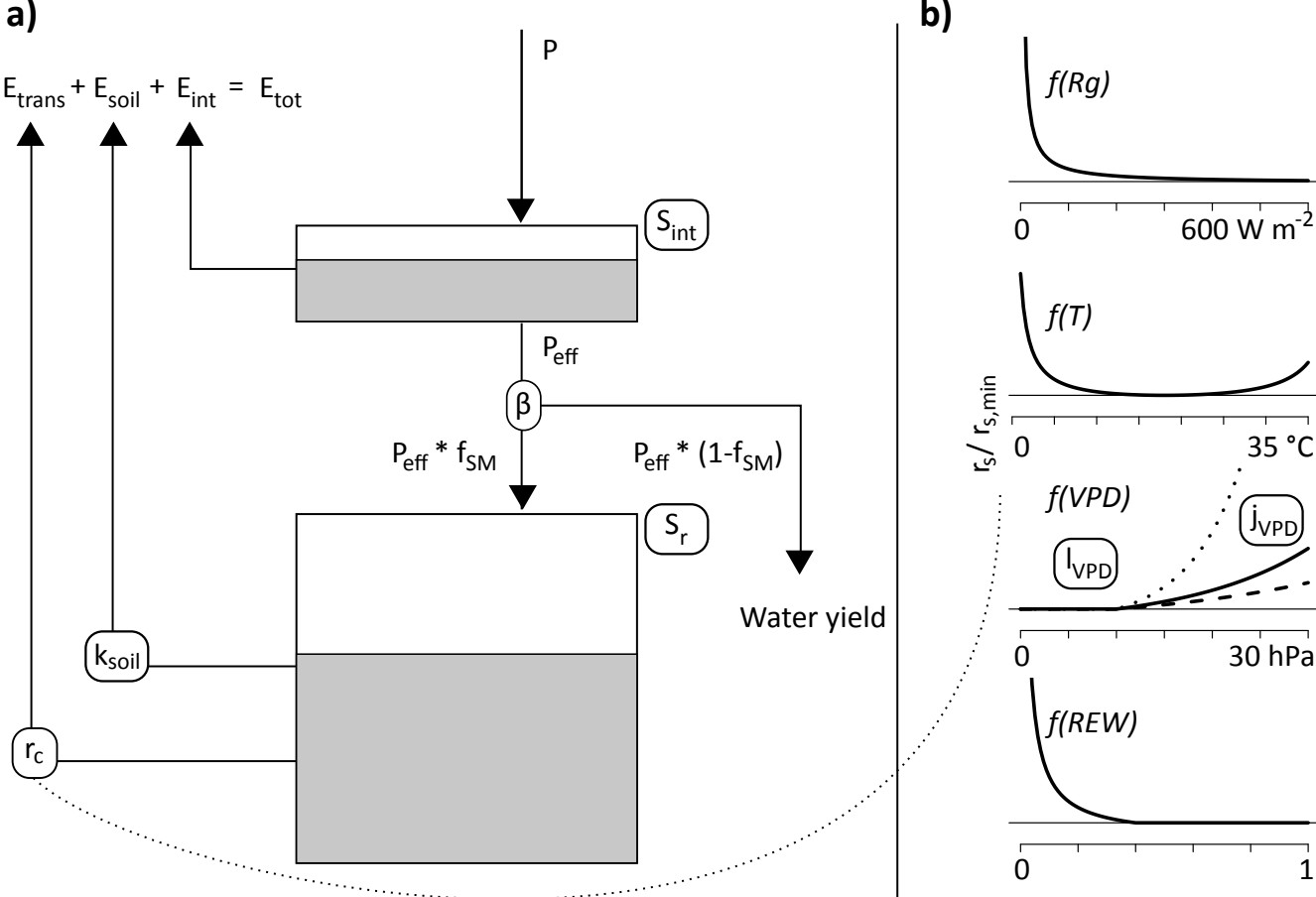

**Figure B1: (a) Schematic representation of the water fluxes in the local water balance model FORHYTM. Incoming precipitation**
**$P$ first fills a canopy interception reservoir of size $S_{int}$, from which water evaporates back to the atmosphere. Precipitation reaching the ground, $P_{eff}$, is split between the rooting zone storage and runoff/groundwater recharge as a function of the parameter $\beta$ and the current filling status of the rooting zone storage. This storage is depleted by soil evaporation and overstory transpiration. Soil evaporation is reduced from its potential value as a function of time since the last rainfall and the parameter $k_{soil}$. Transpiration is controlled by the canopy resistance $r_c$, i.e. stomatal resistance divided by LAI. (b) Relative increase of**
**stomatal resistance as a function of radiation, air temperature, VPD and REW. When all functions are equal to one (optimal conditions), stomatal resistance is equal to $r_{s,min}$.**

A fraction $f_{SM}$ of the water reaching the ground is added to the plant-available soil moisture reservoir $S_{SM}$ as a function of its current filling state and a shape parameter, termed $\beta$:

$$f_{SM} = \left(\frac{S_{SM}}{S_r}\right)^{\beta}. \tag{B1}$$

The remaining fraction of incoming water $(1 - f_{SM})$ is assumed to leave the system as fast runoff or groundwater recharge,

and is not considered further in the model. The reservoir $S_{SM}$ is depleted by canopy transpiration and soil/understory evaporation. The former is controlled by canopy resistance, modeled using a Jarvis-type routine (Jarvis, 1976), whereas the latter is reduced exponentially from its potential value as a function of the number of days without rain (Morillas et al., 2013). The canopy resistance parameterization uses a multiplicative approach, where a minimum stomatal resistance $r_{s,min}$ [s m$^{-1}$] is multiplied with several functions of environmental factors (radiation, temperature, VPD and soil moisture; see Fig.

B1 b)). As long as these factors are not limiting, the corresponding response function has a value of one. The response functions are greater than one (i.e. the resistance is increased) when the corresponding environmental factor has a sub-optimal value. The response functions for radiation, temperature and soil moisture are parameterized following Stewart (1988). For VPD, the model version used here assumes an exponential reduction of stomatal conductance (the inverse of resistance) with increasing VPD. Furthermore, as not all tree species respond to low VPD values, an additional parameter

$l_{vpd}$ [hPa] was introduced, indicating the VPD value above which canopy resistance is affected. The response function for VPD is thus defined as:

$$f_{VPD} = \begin{cases} 1; VPD < l_{VPD} \\ 1/exp\big(j_{VPD} \times (VPD - l_{VPD})\big); VPD \geq l_{VPD} \end{cases}. \tag{B2}$$

Stomatal resistance is then scaled up to canopy scale by dividing by LAI. All calibration parameters of FORHYTM are listed in Table B1. The model also includes a parsimonious snow routine, implemented following Bergström (1992).

All meteorological variables (precipitation, air temperature, VPD, global radiation and wind speed) needed to run FORHYTM are measured at the FLUXNET sites and included in the dataset. The annual maximal leaf area index (LAI) is specified for each site based on literature values (see Table 2). A minimal (winter) value is set based on forest type: 0.2 for deciduous forests, half the maximum value for mixed forests, and LAI is not varied for evergreen forests. The start and end dates of the growing season are calculated as described in Sect. 2.1.4. At the beginning of the growing season, LAI is

linearly increased from its minimum to its maximum value over a period of 30 days. In autumn, LAI is linearly reduced to the minimum over a period of 14 days after the onset of leaf senescence. Site-specific fractional canopy cover was taken from the site description or estimated based on satellite images on Google Earth.

**Table B1: Ranges of the calibration parameters used in this study.**

| Parameter | Units | Meaning | Minimum | Maximum |
|-----------|-------|---------|---------|---------|
| $S_r$ | mm | Size of the plant-available soil moisture reservoir | 30 | 500 |

| | | | | |
|---|---|---|---|---|
| $\beta$ | - | Shape coefficient of the soil moisture recharge function | 1 | 6 |
| $r_{s,min}$ | s m$^{-1}$ | Minimum stomatal resistance | 120 | 1000 |
| $k_{soil}$ | days | e-folding time of the soil evaporation reduction function | 5 | 30 |
| $j_{vpd}$ | - | Exponent of the VPD-induced reduction of stomatal conductance | -0.18 | -0.05 |
| $l_{vpd}$ | hPa | Threshold for stomatal response to VPD | 0 | 20 |
| $k_{int}$ | mm | Link between interception storage and LAI | 1.5 | 4.5 |

Table B2 shows the $KGE$ scores obtained at each site. The $KGE_{evap}$ values in the validation period range from 0.46 to 0.87, and the $KGE_{REW}$ scores range from 0.12 to 0.83. The lowest scores for evaporation were obtained at the Mediterranean sites Roccarespampani and San Rossore, as well as at the montane-Mediterranean site Collelongo. For Collelongo, the bias component of the $KGE_{evap}$ is consistently greater than one in all validation runs (not shown), indicating that the relatively low score at this site is primarily due to a systematic overestimation of $E_{tot}$. The lowest $KGE_{REW}$ was obtained at Loobos (0.12). This site also shows a great uncertainty regarding the value of the optimal $S_r$, as indicated by the large standard deviation. FORHYTM also performed poorly at Roccarespampani, with a $KGE_{REW}$ of 0.32. The calibrated $S_r$ values cover almost the whole parameter range defined in this study and range from 95 (Sodankylä, Renon) to 417 mm (Roccarespampani). The last column of Table B2 shows the parameter importance rank of $S_r$ out of the 7 calibration parameters, as determined by the sensitivity analysis (Sect. 2.3). At all sites, $S_r$ is at least in the third position. This is in line with the results of Speich et al. (2018), who found that $S_r$ had a large influence on long-term water balance predictions. Other important parameters are minimum stomatal resistance $r_{s,min}$ and the VPD response threshold $l_{vpd}$.

**Table B2: Highest $KGE$ scores obtained at each site for calibration (first number) and validation (second number). For the validation period, only scores obtained with calibrated $S_r$ are counted. The last column indicates the importance rank of $S_r$ out of the seven calibration parameters, obtained from the Random Forest-based sensitivity analysis of FORHYTM.**

| Site | Highest $KGE_{evap}$ | Highest $KGE_{REW}$ | Highest $KGE_{AVG}$ | $S_r$ importance rank |
|---|---|---|---|---|
| Vielsalm | 0.75 / 0.87 | 0.88 / - | 0.8 / - | 2 |
| Lägeren | 0.77 / 0.75 | 0.74 / 0.69 | 0.75 / 0.72 | 2 |
| Hainich | 0.8 / 0.8 | 0.58 / - | 0.67 / - | 2 |
| Tharandt | 0.82 / 0.77 | 0.75 / 0.72 | 0.78 / 0.74 | 3 |
| Wetzstein | 0.75 / - | 0.73 / - | 0.72 / - | 3 |

| | | | | |
|---|---|---|---|---|
| Sorø | 0.76 / 0.76 | 0.78 / 0.79 | 0.76 / 0.77 | 2 |
| Hyytiälä | 0.81 / 0.86 | 0.8 / 0.68 | 0.78 / 0.77 | 2 |
| Sodankylä | 0.74 / 0.66 | 0.68 / 0.51 | 0.68 / 0.58 | 1 |
| Le Bray | 0.83 / 0.75 | 0.78 / - | 0.79 / - | 2 |
| Collelongo | 0.85 / 0.55 | 0.89 / - | 0.86 / - | 1 |
| Lavarone | 0.7 / 0.77 | 0.68 / 0.57 | 0.68 / 0.67 | 1 |
| Renon | 0.81 / 0.78 | 0.41 / 0.73 | 0.6 / 0.75 | 2 |
| Roccarespampani | 0.73 / 0.46 | 0.66 / 0.32 | 0.62 / 0.39 | 1 |
| San Rossore | 0.8 / 0.58 | 0.62 / 0.83 | 0.58 / 0.71 | 1 |
| Loobos | 0.84 / 0.75 | 0.63 / 0.12 | 0.71 / 0.44 | 3 |

The time series of the validation runs at Tharandt are shown on Fig. B2 for $E_{tot}$ and for REW. The observations are plotted against the bounds given by the 5th and 95th percentile of the validation runs. For $E_{tot}$, the observations are often close to the lower bound, which indicates a tendency of the model to overestimate $E_{tot}$ at this site. The figure further indicates that the model cannot fully capture the interannual variability, as shown by the overestimation of $E_{tot}$ and of REW in 2006. Another source of disagreement between model and observations is the apparent quick refilling of soil moisture after precipitation events, which is not always reproduced by FORHYTM. Analogous plots for all other stations (except Wetzstein, where no validation was performed) are given on Fig. S3-S15.

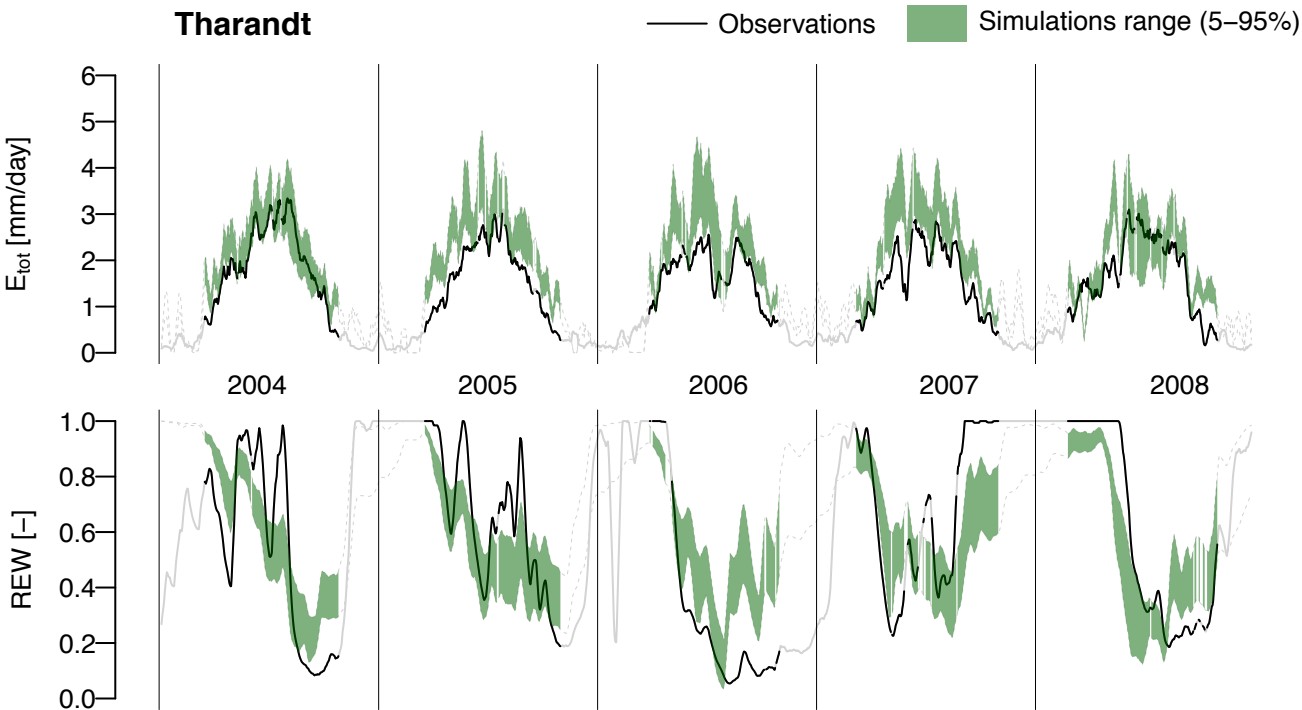

**Figure B2: Time series of total evaporation $E_{tot}$ and relative extractable water REW for the validation period (2004-2008) at Tharandt, comparing the observations with simulations conducted using the parameter sets selected after calibration (see Table 3). The solid line shows the observations, and the dotted lines show the 5 and 95% quantiles of the simulations at each time step. For clarity, the time series are presented here as ten-day moving averages, while the simulations were done with a half-hourly time step.**

At the high-elevation Mediterranean site Collelongo, the relatively low performance during the validation period contrasts with the high calibration efficiency. At this site, the calibration and validation periods (2007-2012 and 1997-2001, respectively) were not immediately contiguous. The disappointing performance in the validation period might therefore be due to changing conditions between the two periods. Indeed, the validation period is characterized by lower $E_{pot}$ (3.49 mm /day, during the calibration period), higher precipitation (2.18 mm/day, versus 1.89), a higher precipitation frequency ($\lambda$=0.14 day$^{-1}$, versus 0.11) and a lower mean intensity ($\alpha$=15.62 mm/event, versus 17.74). As the validation period precedes the calibration period, this indicates a shift towards drier conditions. It is not known whether this change is reflected in vegetation properties, which would further help explain the difference in model performance. In any case, this illustrates the problems with transferring calibrated parameters to new conditions (Bartholomeus et al., 2015). Another particularity of this site is a high spatial heterogeneity of soil depth (Chiti et al., 2010), which is an additional challenge for predicting soil water balance at the scale of the entire site. Furthermore, Hickler et al. (2006) hypothesized that the vegetation has access to groundwater resources at this site, which would lead to an overestimation of the "reservoir" size in the calibration process.

However, physiological indicators of water limitation observed at this site (Scartazza et al., 2013) suggest that the vegetation is at least partially dependent on the water stored in the unsaturated zone.

The good performance of the water balance model at temperate and cold sites suggests that the concept of a bulk $S_r$, defined as the product of soil water holding capacity and effective rooting depth, is an appropriate simplification of reality under these conditions. By contrast, FORHYTM failed to reproduce local water balance properly under Mediterranean climates and on dune soils. This raises the question whether the use of a bulk Sr is appropriate at these locations.

FORHYTM combines three distinct sub-models: the energy partitioning scheme of Guan and Wilson (2009), the Jarvis-type model of canopy resistance, and the soil water balance routine of the hydrological model HBV (Bergström, 1992). Energy partitioning is physically quite well constrained, and the scheme of Guan and Wilson (2009) has been tested under various climates (Lu et al., 2014). On the other hand, previous studies suggest that the two other sub-models may face severe limitations under Mediterranean conditions. For example, Poyatos et al. (2007) calibrated a stand-level evaporation model, including a Jarvis-type parameterization of canopy conductance, in a sub-Mediterranean *Pinus sylvestris* forest. Despite satisfactory calibration efficiency, the model performed poorly during the calibration period. The authors explained this with variations in hydraulic conductance, possibly due to xylem embolism. Recently, Bai et al. (2017) compared different Penman-Monteith based water balance models at Mediterranean eddy covariance sites. Models with a multilayer soil representation performed better than single-layer models. Therefore, under Mediterranean conditions, transpiration may be more sensitive to the vertical distribution of soil moisture and roots. While it is not possible to determine to what extent the canopy resistance or soil water balance submodels contributed to the poor performance of FORHYTM at Mediterranean sites, it is likely that a multi-layer soil model is more appropriate at these sites.

## Data availability

This work used eddy covariance data acquired and shared by the FLUXNET community, including these networks: AmeriFlux, AfriFlux, AsiaFlux, CarboAfrica, CarboEuropeIP, CarboItaly, CarboMont, ChinaFlux, Fluxnet-Canada, GreenGrass, ICOS, KoFlux, LBA, NECC, OzFlux-TERN, TCOS-Siberia, and USCCC. The ERA-Interim reanalysis data are provided by ECMWF and processed by LSCE. The FLUXNET eddy covariance data processing and harmonization was carried out by the European Fluxes Database Cluster, AmeriFlux Management Project, and Fluxdata project of FLUXNET, with the support of CDIAC and ICOS Ecosystem Thematic Center, and the OzFlux, ChinaFlux and AsiaFlux offices.

An R code file containing an implementation of the methods described in this article is provided as a supplement.

## Competing interests

The authors declare that they have no conflict of interest.

**Acknowledgments**

This research was funded by the Swiss National Science Foundation (no. 153544) and the Swiss Federal Office for the Environment (no. 15.0003.PJ / Q104-0149). The authors would like to thank James Kirchner (ETH Zurich) for helpful comments on a previous version of this manuscript. Also, we are grateful to Hubert H. G. Savenije, Andrew J. Guswa and an anonymous reviewer for their valuable feedback, which allowed us to greatly improve the quality of this article.

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
