# Peer review of "Testing an optimality-based model of rooting zone water storage capacity in temperate forests"

_Hydrology and Earth System Sciences, 2017_

## Referee Comment (RC1) · HHG Savenije (Referee) · 11 Jan 2018

This paper presents an optimality model of Guswa (2010) to determine the root zone storage capacity of temperate forests with understory, based on net carbon profit. It then compares it to the root zone storage capacity of a water balance model FORHYTM calibrated on: 1) locally observed total evaporation by eddy covariance; and 2) observed soil moisture.

To me, the value of the paper is not so much that reasonable results are obtained that could potentially be used to predict root zone storage capacity, but rather that by using the net carbon profit approach insight is obtained in what triggers vegetation to evolve

towards a certain storage capacity. From the results obtained in the paper, it is clear that this is not so easy to do. The net carbon profit apparently does not appear to work so well in a Mediterranean climate, and also not on sandy soils. So what we learn form this approach, is that it is apparently not as simple as that.

In my view, the weakness of the approach is that the results are subject to very high uncertainty. Both the optimality model and the conceptual evaporation model are heavily parameterized. Both the soil properties and observed soil moisture suffer from high heterogeneity (both in horizontally and in depth). Also the many coefficients for the phenology, the seasonality and the climate contain substantial uncertainty. The evaporation model, which contains 4 Jarvis-like coefficients for the vegetation in question, has many uncertain parameters as well. So, it is very hard to conclude what we are comparing. We don't know if the mismatches in Fig.7 stem from the optimization model, the water balance model, the heterogeneity in the data (particularly soil moisture and soil properties, but also the fetch of the eddy covariance) or the many parameters and empirical equations used.

This drawback is much less in the climatic water balance approach of Gao et al. (2014) [referred to in the paper], which is primarily data driven, based on the difference between evaporation and precipitation at catchment scale, although also these data have some uncertainty. Gao et al. (2014) compared the estimates of the root zone storage capacity to values of a hydrological model calibrated on the runoff of 329 catchments in the USA and Thailand. Similarly, Wang-Erlandsson et al. (2014) [referred to in the paper], using a similar climatic water balance approach for the entire globe at pixel scale, compared her results with a water balance model similar to FORHYTM, calibrated on a global data set of evaporation based on an energy-balance approach. These two approaches contain much less parameters and less empirical relations, and hence suffer considerably less from uncertainty and provide more reliable estimates of root zone storage capacity at a variety of scales, but – admittedly – present no explanation for the objectives ecosystems apparently try to satisfy.

This is what I think the value of the paper is: providing insight into what triggers ecosystems to evolve towards a certain 'optimal' storage capacity. But then, I would expect the authors to discuss more the relevant objectives of ecosystems under different constraints and also to think of other factors that may determine trade-offs. The authors should realise that getting a model to work for temperate climates is not so difficult. It is far more difficult to predict root zone storage capacity in Mediterranean, semi-arid, or tropical conditions, or under climate change; and this is what we would like to be able to do. Gao et al. (2014) and Wang-Erlandsson et al. (2016) managed to predict root zone storage capacity under present – widely varying – conditions, but without providing insight into what drives the ecosystem to converge to this capacity, besides securing sufficient water to survive (and reproduce).

A few observations, some of them minor:

1. I think that the authors conceptualised the interaction between over- and understory well. I think this is not a problem of the model.

2. I do not recommend using the word "loss". In hydrology there is no loss. Use interception evaporation instead of interception loss, as is done in line 16 of p8. But do it throughout.

3. Please don't use the concocted word evapotranspiration. You managed to avoid it almost everywhere and correctly used the term "total evaporation" or just "evaporation" instead. But it still remains in a few places: line 12 of p8, line 18 of p.9, line 13 of p22.

---

## Referee Comment (RC2) · Anonymous Referee #2 · 18 Jan 2018

**GENERAL COMMENTS**

The reviewed manuscript makes an effort to evaluate an optimality model for zone storage capacity in temperate forests by the use of a hydrological model and eddy covariance data across Europe. Research in root zone storage capacity is important for both improving modelling, and for understanding basic ecological processes. The manuscript is therefore subject-wise appropriate for publication in HESS. However, the research question lacks clear focus and conclusions are made about topics not listed among the research goals, which makes it difficult to comprehend the rationale for the research design as well as the scope of the study. The precise contribution of this

study is not clear, lost somewhere between validity of the FORHYTM itself for application in different regions and the comparison of different parameterisation methods of $S_r$. There is a lack of overview and cross referencing, but an overflow of details in the main manuscript body. I found the document difficult to navigate. Perhaps most worrying, it seems that the authors have not properly isolated the effect of $S_r$ on modelling results and are making speculative claims unsupported by the analyses performed as detailed below. I agree with Prof. Savenije's comment that it would be insightful to provide analyses and explanation of the presumably contrasting mechanisms of root zone storage evolution in boreal and Mediterranean ecosystems. Other general issues include:

- Given the research question stated and some of the conclusions made, I would have expected some more straightforward comparison of evaporation simulation results for the three $S_r$ values: calibrated, Guswa 2008, and Guswa 2010. Please consider providing such figures from such analyses.

- From the Supplementary figures (and the in general relatively large standard deviations shown in Table 5), it appears that a wide range of $S_r$ values can generate high KGE values for most sites, and the same $S_r$ values can also generate vastly different KGE values. This makes me wonder (1) at which sites is $S_r$ of importance for modelling results, and (2) how sensitive the KGE values is to the other calibration parameters listed in Table 4, which the authors also point out on e.g., P24L22. Please consider presenting individual model parameter sensitivity results in a revised manuscript.

- In terms of presentation of methods and results, the authors could do much more to facilitate for the reader. E.g., sites are listed by names or abbreviations, but given that readers will not be taking the effort to memorise the location, vegetation type, and climate of each site, it is very cumbersome for the readers to interpret the results in e.g., Table 5. Please consider adding colour coding or

other visualisation techniques to provide such relevant information.

- Please make relevant cross references when interesting results are discussed. E.g., P25L16, it would have been useful for the reader to be able to cross-check with a figure for exactly how much FORHYTM failed to represent the local water balance under Mediterranean climate.

- Difficulties navigating the different model runs and datasets. Please consider providing an overview table listing the different experiments, and add cross references where appropriate.

- Please discuss the uncertainties in eddy covariance data.

**SPECIFIC COMMENTS**

**Abstract: "the concept of a single rooting zone storage capacity was appropriate at most temperate and cold sites"** This conclusion seems too strong/general. Can e.g., parametrisation, data uncertainty, or model structures not be the reason given the research design and the scope of the performed analyses?

**Abstract: "mismatched were attributed to**...**[]**...**oxygen stress and low soil temperature".** It is not clear to me how the attribution was made. Please consider providing searchable key words that make it easier to locate the related analyses. (I searched for "oxygen" and "attrib" without finding any related analyses).

**Abstract: "Nevertheless, the overall good agreement suggests that this model may be useful for generating estimates of rooting zone storage capacity for both hydrological and ecological applications. Another potential use is the dynamic parameterization of the rooting zone in process-based models, which greatly increases the reliability of transient climate-impact assessment studies."** These are not key conclusions from the study, and rather speculative. I would suggest removing these statements. Introduction: Please clearly state the research questions and

scope. I find the research goal statement too vague at the moment.

**Methods:** Please consider presenting part of the Methods description in Appendix and focus on explaining key features and rationale of the model, with cross-references to Appendix. Notations: Due to the large number of symbols, please consider providing a table in an Appendix section that list all notations.

**P3L4-5: "Yang et al. (2016) identified the approach proposed by Guswa (2008) as the most meaningful from a hydrological and ecological point of view."** This sentence suggests that Yang et al (2016) made a comparison between all aforementioned approaches, which was not the case. The word meaningful is also vague – do you for example mean that this approach yields best performance in both hydrological and ecological modelling or that their approach captures the most major hydrological and ecological drivers of $Z_e$?

**P12-Table3: "LAI".** Do you mean "maximum LAI"? Where is it described how LAI is varied?

**P15-Eq20: "otherwise".** Please consider replacing with formal mathematical expression (VPD=>lvpd?).

**P18-Fig5 caption: "There is a relatively narrow range of $S_r$ leading to Pareto-optimal scores".** The black $S_r$ dots appear to range between approx. 50 and 280 mm. I would be hesitant to refer to this as a narrow range.

**P18-Fig5 caption: "conducted using the optimal parameter sets"** Please be specific and add cross reference. It is not entirely clear which optimal parameter set is considered. The suggested overview table (see General comments) of simulation settings/parameter combinations would be helpful to cross refer to.

**P18 Fig 5 (and SI figures):** Please consider changing the line color and style. At first sight, one might think that the black colors share some common point, which is not the case.

**P21-Fig 7:** Possibly consider collapsing the two subplot columns G08 and G10 into one column, and use colour coding or other visual cues for identifying the model approach used.

**P23:** The cross reference to Fig 4 seems to be wrong.

**P25: "The results of this study suggest that G10 better captures the behavior of forests under energy-limited conditions".** Please consider to add a cross reference. I have difficulties understanding how the analyses and results support this statement.

**P25L17: "suggesting that the use of a bulk $S_r$ is inappropriate at these locations".** I struggle to understand how this claim is supported by the performed analyses. In my view, to be able to make such as claim would require a comparison between a model structure with bulk $S_r$ and a model structure without bulk $S_r$ (e.g., some other structure hypothesised for Mediterranean conditions), and this comparison would need to show that the model structure without bulk $S_r$ performs better than the other one. It seems to me that current analyses only suggest that FORHYTM as a whole does not appear appropriate for modelling evaporation in Mediterranean conditions.

**SI:** Please provide figure numbers and figure captions.

---

## Author Comment (AC1) · 8 Feb 2018

We would like to thank Prof. Savenije for the valuable and constructive feedback on our manuscript. We take the opportunity provided by this forum to address the main concerns raised in this review and offer suggestions for improving the manuscript in the next iteration.

Some of the points have been raised by both reviewers 1 and 2. We address the issues regarding the article structure, thematic focus and research goals/questions mainly in our response to review #1, and concerns regarding sensitivity and uncertainty mainly in our response to review #2.

**Thematic focus and research objective/questions**

One major point of the review concerns the focus of the article. We submitted this manuscript with the intention to test whether Guswa's model yields sensible estimates of Sr in temperate and boreal forests, at the spatial scale of e.g. a forest plot. The aim is to assess whether this parameterization can be implemented in a hydrological or ecohydrological model. As discussed in the Introduction, measurements of rooting depth are scarce and probably not very informative at the spatial scale that we are considering. Therefore, instead of comparing the results of Guswa's model against measurements, we chose to use values obtained by calibration as a reference. As a way to assess the reliability of the calibrated values, a validation of the local water balance model was performed at the stations where the data record was long enough.

One of our main conclusions was that Guswa's model agrees relatively well with the calibrated values in temperate, lowland forests, while mismatches occur e.g. at Mediterranean sites, and at pine sites on coarse soils. However, the review suggests that these are not the most interesting outcomes of the manuscript. Instead, the reviewer sees the value of the paper in the opportunity to gain knowledge on the drivers and processes that influence rooting depth at a given location.

We believe that this point can be reconciled quite well with our initial research goal. Also, one of the main points of criticism of reviewer #2 was the lack of a clear research goal statement. We therefore suggest a reformulation of the research objective as follows:

*The aim of this paper is to assess the suitability of Guswa's model (G10) for implementation in a dynamic hydrological or ecohydrological model. A dynamic Sr parameterization in a hydrological model is suitable if (1) it gives sensible estimates of Sr (or rooting depth) for a given combination of climate, soil and above-ground vegetation, (2) its variations across different climates, soil conditions and vegetation types are physiologically and ecologically justifiable, and (3) the associated uncertainty remains within reasonable bounds. We therefore ask:*
- *How well do the predictions of G10 agree with calibrated values?*
- *How does the sensitivity of G10 to its various inputs vary across sites? Can these variations be explained with physiological and ecological theory?*
- *Given the uncertainty of the inputs to G10, how large is the uncertainty of estimated Sr under different climate/soil/vegetation type combinations?*

The first question is already addressed in the current version of the manuscript. However, as pointed out by reviewers 1 and 2, it is necessary to assess the uncertainty

of the G10 estimates, as well as the FORHYTM results and the calibrated values. We refer to our response to Reviewer #2 for an outline of the sensitivity and uncertainty analyses that we propose, and a brief discussion of preliminary results.

The second question relates to the suggestion of reviewer #1 on the focus of the paper. The different sensitivities of G10 under different conditions can provide insight into the processes that influence rooting depth.

The third question relates directly to the research goal formulated above by assessing (1) whether the uncertainty of G10 is acceptable, and (2) which inputs of G10 are particularly sensitive and should be considered particularly carefully in a dynamic model.

**Discussion of model properties and comparison with alternative approaches**

The answer to these three questions will then form the basis for a discussion of the properties of G10, and how they compare to alternative methods. The water supply-and-demand scheme of Gao et al. and subsequent papers is another candidate for implementation in hydrological models (see e.g. Nijzink et al.), and this is an interesting opportunity to discuss the difference between both approaches. Discussion points include the implied objectives of organisms and/or ecosystems (optimization of C budget vs. resistance to dry spells), the various processes considered in either approach (e.g. seasonal and inter-annual climate variability, role of above-ground vegetation and of soil WHC), and possibly more technical aspects of either approach.
The current manuscript includes a discussion of other possible factors that influence rooting depth (e.g. low soil temperature and oxygen stress). We suggest to include this paragraph in the same subsection as the discussion outlined above.

---

## Author Comment (AC2) · 8 Feb 2018

We would like to thank reviewer #2 for their helpful and constructive comments. We take this opportunity to discuss the main points raised in this review and propose ways to address these concerns in a revised version.

Some of the points have been raised by both reviewers 1 and 2. We address the issues regarding the article structure, thematic focus and research goals/questions mainly in our response to review #1, and concerns regarding sensitivity and uncertainty mainly in our response to review #2.

**Uncertainty of the FORHYTM parameters and the role of Sr**

- Reviewer #2 notes that the sensitivity of FORHYTM to Sr and other parameters is not assessed. We agree that this makes it difficult to interpret the calibrated values and to decide how reliable the calibration procedure is to estimate Sr. Another manuscript of ours (referred to as "in prep.") has been accepted for publication in Environmental Modelling and Software and will be published in volume 102. This article reports on a sensitivity analysis of FORHYTM, assessing the effects of changes in parameters (including soil and canopy properties, as well as the Jarvis parameters) on model outputs (long-term total evaporation, as well as a drought index based on the ratio of actual to potential transpiration). This analysis was conducted at ten sites with contrasting hydro-climatic conditions, including dry inneralpine, temperate lowland and cold subalpine sites. In this study, Sr was among the most influential parameters at all sites, for both output variables.
- To assess whether this also applies to the KGE scores in the present study, we conducted a sensitivity analysis on the calibration runs. At each of the eddy covariance sites, a Random Forest model was fitted to the outputs of the calibration runs, with the seven parameters as predictors, and the KGEavg score as the dependent variable. The Random Forest algorithm provides a measure of variable importance, which allows a ranking of the predictors by their influence on the output. The importance score is based on the increase in model prediction error when the values of a predictor are permutated (see e.g. Liaw and Wiener (2002)) Of the 7 parameters, Sr was the most influential variable at 5 sites, the second most influential at 6 sites and 3$^{rd}$ most influential at 4 sites. While the absolute values of the importance scores vary over several iterations of the RF algorithm, the ranking of the parameters remains stable. Other variables of importance are RSmin and l_vpd. The following figure shows the variable importance scores at three stations, with Sr in the first, second and third position:

[Figure]

**Sensitivity and uncertainty of the G10 model**

- We also agree that it is important to assess the sensitivity of the G10 model to its inputs (climate statistics, physiological parameters, LAI, soil WHC). We have started to assess the effect of variations in physiological parameters in Sections 2.1.4 and 3.3. However, we now feel that a more formal sensitivity and uncertainty analysis would be much more informative.
- We have generated 2000 estimates of Sr at each station, with perturbations of all parameters by up to 20%. The parameters include (1) plant physiological parameters for trees and grass, (2) climate statistics, (3) site characteristics such as LAI and soil WHC. In addition, the start and end of the growing season were also shifted back or forward by up to 10 days (which, in turn, also affects the climate statistics calculated over the growing season). Sampling was again done with the Latin Hypercube method.
- The resulting standard deviations of Sr ranged between 18 and 59 mm across sites. The spread, however, is much larger. Distributions of Sr tend to have long

tails, with a few extreme values. It will be useful to determine which parameter combinations lead to these extremes.

- We also applied a Random Forest model to determine variable importance, in a similar manner as for the FORHYTM calibration runs. Preliminary results indicate for example that the parameter values for grass have little importance at all sites. The sensitivity rankings further suggest geographical differences, with e.g. LAI being more sensitive at the Mediterranean sites than elsewhere. While these preliminary results are promising, it might be necessary to define more specific uncertainty bounds than 20% for each parameter. This will be based on a brief discussion of the various sources of uncertainty associated with each parameter.

**Uncertainty of the eddy covariance and soil moisture data**

- Although we briefly mention some of the issues of eddy covariance and soil moisture data, we agree that it is worthwhile to discuss these in more detail, and to put the FORHYTM results and calibrated values in this context. We plan to include a paragraph or subsection in the discussion where we briefly summarize the recent literature published by the EC community on these issues (regarding e.g. random and systematic measurement errors, as well as spatial heterogeneity), and relate them to the analysis in this article.

- The review points out several statements where it is not clear to what extent they are supported by the analyses. We will carefully review each of the statements and either clarify the link to the analysis or reformulate them to avoid making unsubstantiated claims.

**Article structure and readability**

- We also appreciate the feedback on the structure and readability of the article. In our response to reviewer #1, we have outlined a possible reformulation of the research goals and questions. We also propose to rearrange the structure, so that only the parts that directly relate to the research questions are kept in the main text, and accessory parts are shifted to the appendix. For example, the validation of FORHYTM is useful to estimate the reliability of model results and of the calibrated values (in addition to the sensitivity analysis outlined above), but does not directly relate to the research questions and can be moved to the appendix. Also, we suggest to move the section on numerical approximation to the Appendix. For the description of FORHYTM, we can now refer to the Environ. Modell. Softw. article. We believe that this will further clarify the goals of the article and enhance readability.
- We will also carefully consider the points on cross-referencing, and presentation of the results. We will add color to the figures where appropriate.

Liaw, A. and M. Wiener (2002): *Classification and Regression by randomForest*. R News 2(3), 18-22.

---

## Author Comment (AC3) · 28 Feb 2018

**Responses to specific comments of Reviewer #1**

I do not recommend using the word "loss". In hydrology there is no loss. Use interception evaporation instead of interception loss, as is done in line 16 of p8. But do it throughout.

Agreed. We will replace the three occurrences of "interception loss" with "interception evaporation".

Please don't use the concocted word evapotranspiration. You managed to avoid it almost everywhere and correctly used the term "total evaporation" or just "evaporation" instead. But it still remains in a few places: line 12 of p8, line 18 of p.9, line 13 of p22.

We also agree with this comment and will replace "evapotranspiration" with "evaporation".

**Responses to specific comments of Reviewer #2**

Abstract: "the concept of a single rooting zone storage capacity was appropriate at most temperate and cold sites" This conclusion seems too strong/general. Can e.g., parametrisation, data uncertainty, or model structures not be the reason given the research design and the scope of the performed analyses?

Indeed, this statement is outside the scope of the research question, and not necessarily supported by the analysis.

Abstract: "mismatched were attributed to...[]...oxygen stress and low soil temperature". It is not clear to me how the attribution was made. Please consider providing searchable key words that make it easier to locate the related analyses. (I searched for "oxygen" and "attrib" without finding any related analyses).

The term "attribution" might be misleading here – this statement refers to a possible (but untested) explanation for the mismatch at high-elevation spruce sites. This is perhaps given too much weight in the abstract. In a revised version, we will simply state that in some situations, factors other than carbon uptake may control rooting depth, such as cavitation risk, low soil temperature or oxygen stress.

Abstract: "Nevertheless, the overall good agreement suggests that this model may be useful for generating estimates of rooting zone storage capacity for both hydrological and ecological applications. Another potential use is the dynamic parameterization of the rooting zone in process-based models, which greatly increases the reliability of transient climate-impact assessment studies." These are not key conclusions from the study, and rather speculative. I would suggest removing these statements.

As we write in our comment AC1 (Response to reviewer #1), assessing the potential of the G10 method for implementation in a dynamic model was our primary motivation for this study. As we have reformulated our research goal and questions, it makes more sense to mention this at the beginning of the abstract. In our revised research questions

(see AC1), we state the criteria by which we assess the suitability of the G10 model for this purpose. Therefore, the sentences quoted above can be removed, and replaced by more specific statements on the research questions.

**P3L4-5: "Yang et al. (2016) identified the approach proposed by Guswa (2008) as the most meaningful from a hydrological and ecological point of view."** This sentence suggests that Yang et al (2016) made a comparison between all aforementioned approaches, which was not the case. The word meaningful is also vague – do you for example mean that this approach yields best performance in both hydrological and ecological modelling or that their approach captures the most major hydrological and ecological drivers of Ze?

We agree- this formulation is potentially misleading. We will adapt it accordingly. The word "meaningful" (in the second sense) provides a link to the points discussed in RC1 and AC1 (comparison with data-driven approaches)

**P12-Table3: "LAI".** Do you mean "maximum LAI"? Where is it described how LAI is varied?

This is indeed the maximum LAI. We assume no inter-annual variation of maximum LAI. The description of intra-annual variations of LAI in deciduous and mixed stands is indeed incomplete, and we will add it in a revised version.

**P18-Fig5 caption: "There is a relatively narrow range of Sr leading to Paretooptimal scores".** The black Sr dots appear to range between approx. 50 and 280 mm. I would be hesitant to refer to this as a narrow range.

Agreed.

**P18-Fig5 caption: "conducted using the optimal parameter sets"** Please be specific and add cross reference. It is not entirely clear which optimal parameter set is considered. The suggested overview table (see General comments) of simulation settings/parameter combinations would be helpful to cross refer to.

Agreed – we will include a table and reformulate this sentence accordingly.

**P18 Fig 5 (and SI figures):** Please consider changing the line color and style. At first sight, one might think that the black colors share some common point, which is not the case.

We agree that this will improve the legibility of these figures.

**P21-Fig 7:** Possibly consider collapsing the two subplot columns G08 and G10 into one column, and use colour coding or other visual cues for identifying the model approach used.

This could also be helpful. Furthermore, as we will replace the rough parameter sensitivity analysis with a more formal uncertainty analysis (see AC2), the two bottom panels can be removed.

P23: The cross reference to Fig 4 seems to be wrong.

Yes, the correct reference is Fig. 7.

**P25: "The results of this study suggest that G10 better captures the behavior of forests under energy-limited conditions".** Please consider to add a cross reference. I have difficulties understanding how the analyses and results support this statement.

This should refer to the discussion in the first paragraph of Section 4.2. However, instead of a cross-reference, it is perhaps preferable to merge these two paragraphs (first paragraph of Sect 4.2 and last paragraph of Sect 4.3).

**P25L17:** "suggesting that the use of a bulk Sr is inappropriate at these locations". I struggle to understand how this claim is supported by the performed analyses. In my view, to be able to make such as claim would require a comparison between a model structure with bulk Sr and a model structure without bulk Sr (e.g., some other structure hypothesised for Mediterranean conditions), and this comparison would need to show that the model structure without bulk Sr performs better than the other one. It seems to me that current analyses only suggest that FORHYTM as a whole does not appear appropriate for modelling evaporation in Mediterranean conditions.

We suggest replacing this statement with the following :

"By contrast, FORHYTM failed to reproduce local water balance properly under Mediterranean climates and on dune soils. This raises the question whether the use of a bulk Sr is appropriate at these locations.

FORHYTM combines three distinct sub-models: the energy partitioning scheme of Guan and Wilson (2009), the Jarvis-type model of canopy resistance, and the soil water balance routine of the hydrological model HBV (Bergström, 1992). Energy partitioning is physically quite well constrained, and the scheme of Guan and Wilson (2009) has been tested under various climates (Lu et al., 2014). On the other hand, previous studies suggest that the two other sub-models may face severe limitations under Mediterranean conditions. For example, Poyatos et al. (2007) calibrated a stand-level evaporation model, including a Jarvis-type parameterization of canopy conductance, in a sub-Mediterranean *Pinus sylvestris* forest. Despite satisfactory calibration efficiency, the model performed poorly during the calibration period. The authors explained this with variations in hydraulic conductance, possibly due to xylem embolism. Recently, Bai et al. (2017) compared different Penman-Monteith based water balance models at Mediterranean eddy covariance sites. Models with a multilayer soil representation performed better than single-layer models. Therefore, under Mediterranean conditions, transpiration may be more sensitive to the vertical distribution of soil moisture and roots. While it is not possible to determine to what extent the canopy resistance or soil water balance submodels contributed to the poor performance of FORHYTM at Mediterranean sites, it is likely that a multi-layer soil model is more appropriate at these sites."

To avoid redundancy, the first paragraph of Section 4.1, where the Bai et al. (2017) article is discussed, should be simplified accordingly.

**Additional references**

- Lu, H., Liu, T., Yang, Y., Yao, D., 2014. A Hybrid Dual-Source Model of Estimating Evapotranspiration over Different Ecosystems and Implications for Satellite-Based Approaches. Remote Sens. 6, 8359–8386. https://doi.org/10.3390/rs6098359
- Poyatos, R., Villagarcía, L., Domingo, F., Piñol, J., Llorens, P., 2007. Modelling evapotranspiration in a Scots pine stand under Mediterranean mountain climate using the GLUE methodology. Agricultural and Forest Meteorology 146, 13–28. https://doi.org/10.1016/j.agrformet.2007.05.003

---

## Referee Report (RR1)

**Review of HESS-2017-723**
Speich, Lischke, and Zappa
Testing an optimality-based model of rooting zone water storage capacity in temperate forests

Review by Andrew J. Guswa, Smith College, aguswa@smith.edu

In this manuscript, the authors use eddy covariance and soil-moisture measurements from fifteen European sites to calibrate a water-balance model (FORHYTM). Root-zone storage, one of the calibrated parameters, is then compared with predictions of root-zone storage (Sr) from two theoretical models - G08, G10 – that presuppose optimality for plant-roots with respect to carbon. In the interest of transparency, I acknowledge that I am the author of the papers that originally presented those theoretical models (Guswa, 2008; Guswa, 2010).

I have seen only the revised version of the paper, and the authors seem to have addressed the comments of the previous reviewers. With respect to the results and discussion, I offer two additional comments:

1. The Mediterranean sites (IT-Sro, IT-Ro2, and IT-Col) are three sites for which the optimality-based Sr was calculated to be far less than the calibrated. Thus, it may be that "net carbon profit apparently does not appear to work so well" at those sites (as mentioned by Dr. Savenije, Reviewer #1). The authors mention that the difference between calibrated and optimal may be attributed to the fact that these sites have lower performance of the water-balance model (so the calibrated Sr values are subject to uncertainty) and that these sites have younger vegetation and may also be affected by the presence of shallow groundwater.

An additional explanation for the difference could be the inadequacy of the precipitation model; indeed, in a Mediterranean climate, the intermittency of events during the growing season is likely not as important as the lower-frequency signal of wet and dry seasons. Such seasonality is not well represented by the Poisson model of rain arrivals; it may need to be approximated by using a much lower frequency of events (see section 2.2.1 and Figure 4 in Guswa, 2008).

2. Another issue that may be confounding the results is the method by which the authors separate the root zone into understory and overstory components, which are then summed to get the total root-zone storage capacity. My understanding of the way that this is accomplished is as follows:

a. the evaporative demand (Epot) is partitioned into an overstory and understory evaporation based on LAI.

b. a wetness index for the overstory (W,o) and the understory (W,u) is computed by Peff/Tpot,o and Peff/Tpot,u, respectively.  The authors acknowledge that they are ignoring competition for Peff between the overstory and the understory.

c. A root-zone storage (depth) is computed for the overstory (Sr,o) and understory (Sr,u) by applying the optimality models G8 and G10, using W,o and W,u, along with vegetation parameters.

d. The total root-zone storage is computed by summing Sr,o and Sr,u

The challenge with this approach is that the optimal root depth (or root-zone storage) for the overstory and understory is based upon wetness indices (W,o and W,u) that are not necessarily representative of the climate.  By partitioning the energy between overstory and understory, but not partitioning the water, the wetness indices will be larger than they should be.  In reality, not all of Peff would be available to the understory, nor would it all be available for the overstory.  However, the formulation in the paper computes the wetness index presuming that all of the precipitation is available to both the understory and the overstory.  Thus, the optimality models, G8 and G10, would compute a root depth for the overstory and understory based on a wetness index that is too wet.  This, combined with the non-linearity of root-depth as a function of wetness, means that the total root-zone storage will be a strong function of LAI, not only because LAI dictates the partitioning of energy between overstory and understory but because it also changes the wetness index.

As an example, using parameter values that are similar to those used in the paper, the table below shows how total root-zone storage changes as a function of LAI, even when the vegetation parameters are the same in the understory and overstory.  For example, with an LAI of 2, the energy is partitioned approximately equally between the understory and overstory, leading to W values of approximately 2.  When LAI,o is very large or very small, however, the wetness index for the dominant vegetation type is closer to 1.  This leads to root-depths that vary from 150 mm to 240 mm, revealing a potential artifact of the method.

| Andrew J. Guswa | | Peff = | 2.5 mm/day | | | theta = | 0.15 | | |
|---|---|---|---|---|---|---|---|---|---|
| Review of HESS-2017-723 | | Epot = | 3 mm/day | | | alpha = | 15 mm | | |
| 15 May 2018 | | k = | 0.5 | | | theta/alpha = | 0.01 mm$^{-1}$ | | |
| | | | | | | Beta/Tpot = | 1.67E+02 day/mm | | |

| LAI,o | Tpot,o | Tpot,u | W,o | W,u | Beta,o | Beta,u | Sr,o | Sr,u | S,total |
| | mm/day | mm/day | - | - | - | - | mm | mm | mm |
|---|---|---|---|---|---|---|---|---|---|
| 0 | 0.00 | 3.00 | | 0.83 | | 500.0 | 0 | 230 | 230 |
| 1 | 0.89 | 1.82 | 2.82 | 1.37 | 147.6 | 303.3 | 43 | 140 | 183 |
| 2 | 1.42 | 1.10 | 1.76 | 2.27 | 237.0 | 183.9 | 86 | 58 | 144 |
| 3 | 1.75 | 0.67 | 1.43 | 3.73 | 291.3 | 111.6 | 128 | 30 | 158 |
| 5 | 2.07 | 0.25 | 1.21 | 10.15 | 344.2 | 41.0 | 189 | 9.5 | 199 |
| 10 | 2.23 | 0.02 | 1.12 | 123.68 | 372.5 | 3.4 | 233 | 0.7 | 234 |
| 20 | 2.25 | 0.00 | 1.11 | | 375.0 | | 238 | 0 | 238 |

Beta is the beta parameter from Guswa, 2008, which combines vegetation parameters with climate and soil

Of course, I may be misinterpreting what the authors have done. I recommend that the paper be returned to the authors for revision, comment, or correction with respect to partitioning the root-zone storage between the overstory and understory. I would be happy to answer questions that the authors might have.

---

## Author Response (AR2)

**Comments from the Editor:**

**Dear Authors:**

**From the comments received during the second round of evaluation I still see that your paper should require a few changes (if you agree with them). Ref.#1, albeit happy somehow with your revision, has still evaluated the scientific significance and quality of your contribution as rather poor. It would help if an improvement might be done by trying perhaps to go more in-depth about the main drivers leading toward the attainment of a certain level of storage capacity. Moreover, Ref.#2 (prof. Guswa) have provided some useful comments and direction for improvements, which I suggest the authors should follows. Please, work a bit more on your last version of the paper and submit a new paper together with the relevant responses.**

We are glad to get another opportunity to improve our manuscript. In addition to the changes suggested by Prof. Guswa (see below), we now also discuss how the G08/G10 models might behave under climate change, in light of their properties and of the analysis conducted in this paper (Sect. 4.3, p. 29 l. 10 and following). We hope that you will find these modifications useful.
* * *
**Comments from Reviewer #2:**

**In this manuscript, the authors use eddy covariance and soil-moisture measurements from fifteen European sites to calibrate a water-balance model (FORHYTM). Root-zone storage, one of the calibrated parameters, is then compared with predictions of root-zone storage (Sr) from two theoretical models - G08, G10 – that presuppose optimality for plant-roots with respect to carbon. In the interest of transparency, I acknowledge that I am the author of the papersthat originally presented those theoretical models (Guswa, 2008; Guswa, 2010). I have seen only the revised version of the paper, and the authors seem to have addressed the comments of the previous reviewers. With respect to the results and discussion, I offer two additional comments:**

We would like to thank Prof. Guswa for his comments on our manuscript. The two points raised in this review give us the opportunity to improve our methodology and strengthen our discussion. In the following, we specifically address both of these points:

**1. The Mediterranean sites (IT-Sro, IT-Ro2, and IT-Col) are three sites for which the optimality-based Sr was calculated to be far less than the calibrated. Thus, it may be that "net carbon profit apparently does not appear to work so well" at those sites (as mentioned by Dr. Savenije, Reviewer #1). The authors mention that the difference between calibrated and optimal may be attributed to the fact that these sites have lower performance of the water-balance model (so the calibrated Sr values are subject to uncertainty) and that these sites have younger vegetation and may also be affected by the presence of shallow groundwater.**

**An additional explanation for the difference could be the inadequacy of the precipitation model; indeed, in a Mediterranean climate, the intermittency of events during the growing season is likely not as important as the lower frequency signal of wet and dry seasons. Such seasonality is not well represented by the Poisson model of rain arrivals; it may need to be approximated by using a much lower frequency of events (see section 2.2.1 and Figure 4 in Guswa, 2008).**

It seems indeed that the representation of precipitation is a plausible explanation for the low values obtained at the Mediterranean sites. In the new version of the manuscript, we describe the issue in the discussion (Sect. 4.2.3, p. 27). We test the suggested parameterization from Guswa (2008) at the four Mediterranean sites. This is described in Sect. 4.2.3 (p. 27, l. 15 and following).

**2. Another issue that may be confounding the results is the method by which the authors separate the root zone into understory and overstory components, which are then summed to get the total root-zone storage capacity. My understanding of the way that this is accomplished is as follows:**
**a. the evaporative demand (Epot) is partitioned into an overstory and understory evaporation based on LAI.**
**b. a wetness index for the overstory (W,o) and the understory (W,u) is computed by Peff/Tpot,o and Peff/Tpot,u, respectively. The authors acknowledge that they are ignoring competition for Peff between the overstory and the understory.**
**c. A root-zone storage (depth) is computed for the overstory (Sr,o) and understory (Sr,u) by applying the optimality models G8 and G10, using W,o and W,u, along with vegetation parameters.**
**d. The total root-zone storage is computed by summing Sr,o and Sr,u**

**The challenge with this approach is that the optimal root depth (or root-zone storage) for the overstory and understory is based upon wetness indices (W,o and W,u) that are not necessarily representative of the climate. By partitioning the energy between overstory and understory, but not partitioning the water, the wetness indices will be larger than they should be. In reality, not all of Peff would be available to the understory, nor would it all be available for the overstory.**
**However, the formulation in the paper computes the wetness index presuming that all of the precipitation is available to both the understory and the overstory. Thus, the optimality models, G8 and G10, would compute a root depth for the overstory and understory based on a wetness index that is too wet. This, combined with the non-linearity of root-depth as a function of wetness, means that the total root-zone storage will be a strong function of LAI, not only because LAI dictates the partitioning of energy between overstory and understory but because it also changes the wetness index.**
**As an example, using parameter values that are similar to those used in the paper, the table below shows how total root-zone storage changes as a function of LAI, even when the vegetation parameters are the same in the understory and overstory. For example, with an LAI of 2, the energy is partitioned approximately equally between the understory and overstory, leading to W values of approximately 2. When LAI,o is very large or very small, however, the wetness index for the dominant vegetation type is closer to 1. This leads to root-depths that vary from 150 mm to 240 mm, revealing a potential artifact of the method.**

Andrew J. Guswa
Review of HESS-2017-723
15 May 2018

| | | | |
|---|---|---|---|
| Peff = | 2.5 mm/day | theta = | 0.15 |
| Epot = | 3 mm/day | alpha = | 15 mm |
| k = | 0.5 | theta/alpha | $0.01$ mm$^{-1}$ |
| | | Beta/Tpot = | 1.67E+02 day/mm |

| LAI,o | Tpot,o mm/day | Tpot,u mm/day | W,o - | W,u - | Beta,o - | Beta,u - | Sr,o mm | Sr,u mm | S,total mm |
|---|---|---|---|---|---|---|---|---|---|
| 0 | 0.00 | 3.00 | | 0.83 | | 500.0 | 0 | 230 | 230 |
| 1 | 0.89 | 1.82 | 2.82 | 1.37 | 147.6 | 303.3 | 43 | 140 | 183 |
| 2 | 1.42 | 1.10 | 1.76 | 2.27 | 237.0 | 183.9 | 86 | 58 | 144 |
| 3 | 1.75 | 0.67 | 1.43 | 3.73 | 291.3 | 111.6 | 128 | 30 | 158 |
| 5 | 2.07 | 0.25 | 1.21 | 10.15 | 344.2 | 41.0 | 189 | 9.5 | 199 |
| 10 | 2.23 | 0.02 | 1.12 | 123.68 | 372.5 | 3.4 | 233 | 0.7 | 234 |
| 20 | 2.25 | 0.00 | 1.11 | | 375.0 | | 238 | 0 | 238 |

Beta is the beta parameter from Guswa, 2008, which combines vegetation parameters with climate and soil

**Of course, I may be misinterpreting what the authors have done. I recommend that the paper be returned to the authors for revision, comment, or correction with respect to partitioning the root-zone storage between the overstory and understory. I would be happy to answer questions that the authors might have.**

The first paragraph correctly summarizes the partitioning between overstory and understory as we implemented it in the original version. We agree with the reviewer that this method is flawed, as the resulting wetness indices become too high. We propose a simpler alternative to estimate rooting zone storage for the overstory and understory. This approach is described in Section 2.2, starting on p. 5. Figure 1 was slightly modified.

The $S_r$ estimates obtained with the new method differ somewhat from those of the previous version (Table 4, Fig. 5), but the general pattern is still similar. All relevant tables and figures (Table 4, Fig. 5-8) were updated and show the results obtained with the new parameterization. Also, the specific T_pot and wetness indices for understory and overstory are no longer used, and are therefore no longer reported (Table 5, Fig. 7).

As a result, the GFor models are less sensitive to changes in LAI (Fig. 6, 7). We have adapted the discussion accordingly in Section 4.2.2.

[revised manuscript text omitted]